# Higher tree diversity increases soil microbial resistance to drought

Lauren M. Gillespie [1✉], Nathalie Fromin [1], Alexandru Milcu [1,2], Bruno Buatois[1], Clovis Pontoizeau[1] & Stephan Hättenschwiler[1]

Predicted increases in drought frequency and severity may change soil microbial functioning. Microbial resistance and recovery to drought depend on plant community characteristics, among other factors, yet how changes in plant diversity modify microbial drought responses is uncertain. Here, we assessed how repeated drying-rewetting cycles affect soil microbial functioning and whether tree species diversity modifies these effects with a microcosm experiment using soils from different European forests. Our results show that microbial aerobic respiration and denitrification decline under drought but are similar in single and mixed tree species forests. However, microbial communities from mixed forests resist drought better than those from mono-specific forests. This positive tree species mixture effect is robust across forests differing in environmental conditions and species composition. Our data show that mixed forests mitigate drought effects on soil microbial processes, suggesting greater stability of biogeochemical cycling in mixed forests should drought frequency increase in the future.

[1] Centre d'Ecologie Fonctionnelle et Evolutive (CEFE), CNRS, UMR 5175, Université de Montpellier, Université Paul Valéry, EPHE, IRD, Montpellier, France. [2] Ecotron Européen de Montpellier, CNRS, UPS, 32980 Montferrier-sur-Lez, France. ✉email: lgillespie155@gmail.com

Climate change models predict increased drought frequency and severity in the Americas, southern Europe, southern and central Africa, Australia, and southeast Asia in the twenty-first century[1,2], which may have far reaching consequences for ecosystem stability and functioning[3,4]. Tree mortality and forest dieback are also projected to increase in association with climate change induced drought, potentially leading to decreased forest carbon storage[1]. In order to understand how increasing drought affects ecosystems, it is imperative to understand how soil microbial communities respond to climate change-induced shifts in soil moisture dynamics, due to their critical role in ecosystem functioning[5,6]. Although soil microbial communities are regularly exposed to drying-rewetting (DRW) cycles in most ecosystem types[7], increasing drought duration and DRW cycle frequency may induce shifts in microbial community composition, biomass, and activity[8–10], ultimately affecting biogeochemical cycling rates[7,11]. Changes in biogeochemical cycling have major impacts on soil carbon dynamics[12], nutrient availability[5], greenhouse gas fluxes between soils and the atmosphere[13], and water soluble compounds leaching from the system[14,15].

Soil microbial resistance (defined as the degree to which microbial activity changes during a disturbance[16]) and recovery (defined as the degree that the activity recovers after the disturbance[16]) are key properties of microbial communities and how they respond to increasing DRW severity and frequency. These properties appear tied to soil parameters and resource availability[17,18]. Soil resource availability, particularly carbon (C) and nitrogen (N) concentrations, determines microbial community composition and structure[19] and also regulates microbial ability to produce molecules, such as osmolytes, for protection against rapid osmotic changes or to recover post-drought[7]. Distinct microbial communities are likely to differ in their responses to drought and changing DRW cycles. For example, fungi-dominant communities may better tolerate drought events than bacteria-dominated microbial communities[20,21]. Gaseous C and N fluxes (e.g., $CO_2$ and $N_2O$) or dissolved compounds susceptible to being lost by leaching (e.g., dissolved organic carbon and dissolved nitrogen) can thus be useful proxies for microbial activity and potential resource loss changes due to DRW. Soil properties are heavily influenced by plant community composition and diversity through root exudation, litter decomposition, and mycorrhizal associations[5,17,19,22,23]. Plant effects on soil properties can be seen in the soil legacy long after the plant community had changed or dissapeared[24,25]. Root traits of different plant species associated with different resource use strategies (e.g., acquisitive or conservative) can directly influence the plant–soil microbial interactions through variance in exudate quality and/or quantity[26] due to either dominance effects (mass ratio theory[27]) or varying functional diversity (higher diversity leading to improved resource partitioning[28]). Resource-based effects on microbial resistance and/or recovery could then act through increased physiological performance of specific taxa or through increased microbial diversity, with a higher chance of more resistant or faster recovering taxa being present in more diverse communities[29]. However, the existing data on plant diversity effects on the response of soil microbial community to stresses are nonconclusive[17,19,22,30], particularly regarding effects of more frequent and/or intense DRW events. Limited evidence indicates that the higher plant diversity may promote microbial resistance and resilience to drought[17,31]. However, the generality of these responses beyond site-specific conditions, and the relative importance of plant functional diversity and of plant trait dominance within a stand compared to the commonly predominant soil effects, which vary widely among ecosystems, remains unknown.

In a microcosm experiment with soil from four mature, natural forests in various soil and climatic conditions (including a total of 13 tree species and 34 different species combinations) along a latitudinal gradient stretching across Europe (Supplementary Table 1, Supplementary Fig. 1), we assessed how tree species mixing (monospecific vs. 3-species mixed stands) affects microbial responses to repeated DRW cycles. Specifically, we asked how DRW cycles influence soil microbial driven carbon and nitrogen cycling by measuring aerobic respiration ($CO_2$ fluxes) and denitrification ($N_2O$ fluxes when $N_2$ transformation is impeded) and potential soluble carbon and nitrogen leaching from soils originating from boreal, temperate, and Mediterranean forests across Europe, and whether these effects are modified by tree species mixing. Soil parameters and absorptive root traits, either calculated as functional diversity or community weighted means (tied to trait dominance), were also included in analyses as potential significant factors in microbial responses. We hypothesize that although DRW cycles have generally negative effects on microbial functioning, mixed tree species forests mitigate these negative effects by increasing microbial resistance and/or recovery. Results showed that $CO_2$ and $N_2O$ fluxes decline under drought but were similar in soils from single and mixed tree species forests. However, soil microbial communities from mixed forests resisted drought better than those from monospecific forests. This positive tree species mixture effect was robust across forests differing in environmental conditions and species composition and suggests potentially more stable biogeochemical cycling in mixed forests with future climate change induced drought.

## Results

**$CO_2$ and $N_2O$ fluxes.** $CO_2$ and $N_2O$ fluxes, reflecting microbial aerobic respiration and denitrification activity respectively, did not differ between control and DRW treatments at the beginning of the experiment (Figs. 1a and 2a). Compared to control conditions, the $CO_2$ fluxes were lower following drought (−48% and −61% on average after the first and second drought, respectively, Fig. 1b, d) and higher 7 days after rewetting (+24% and +26% after the first and the second rewetting, respectively Fig. 1c, e) for both DRW cycles (significant DRW treatment × experimental period interaction; Table 1, Fig. 1). Overall, $CO_2$ fluxes were lower after the second DRW cycle compared to the first for both drought (control: −15%, DRW: −39%) and rewetting periods (control: −37%, DRW: −36%). Tree species number did not explain $CO_2$ flux variability (Table 1). In addition to tree species mixing, trees can leave important footprints in the soil via their root characteristics (e.g., root chemistry, morphology, and mycorrhiza), potentially altering exudate quantity and quality. We considered such root trait effects on gas fluxes by evaluating how trait dominance (community weighed mean, CWM) and functional diversity (functional dispersion, FDis) of a number of key absorptive root traits (Supplementary Table 2, Supplementary Fig. 2) influence microbial activity. Indeed, there was a significant CWM root trait legacy effect on $CO_2$ fluxes, with higher fluxes associated with more conservative root traits (larger diameter and higher tissue and length density; Table 1). When $CO_2$ fluxes were cumulated over the entire experiment, they were 16% lower in the DRW treatment compared to the control (Fig. 1f). The cumulated $CO_2$ respired from mono-specific stand soils did not differ from mixed stands irrespective of DRW treatment (Table 1). There was the same CWM root trait effect on cumulative $CO_2$ fluxes with higher cumulative fluxes with the conservative root traits mentioned above (Table 1, Supplementary Fig. 2).

$N_2O$ fluxes changed similarly in response to drought as $CO_2$ fluxes (Fig. 2b, d). Compared to control conditions, the $N_2O$ fluxes decreased during drought in both DRW cycles, i.e.,

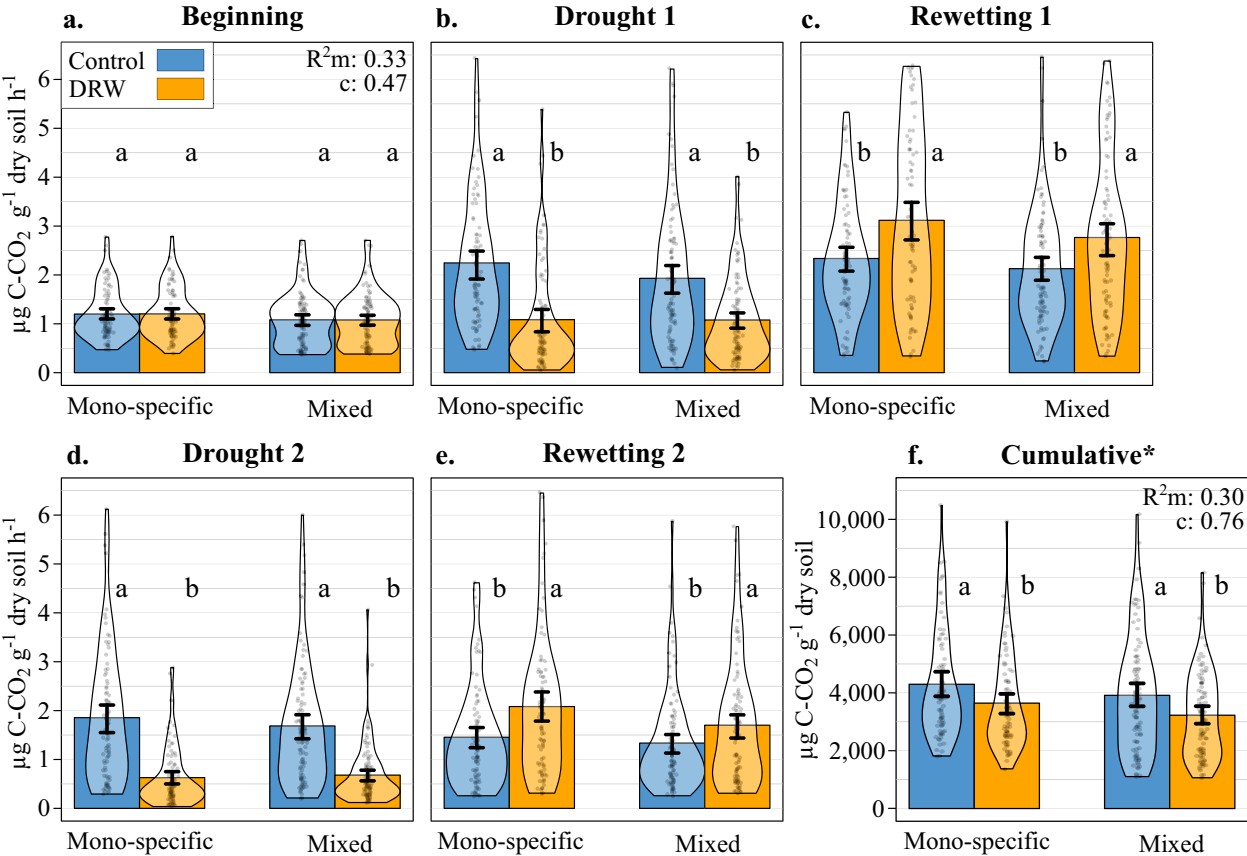

**Fig. 1 CO$_2$ flux and cumulative fluxes (µg C–CO$_2$ g$^{-1}$ dry soil h$^{-1}$) over the five key DRW experimental periods (microcosms: $n = 90$ monospecific × DRW, $n = 90$ mono-specific × control, $n = 102$ mixed × DRW, $n = 102$ mixed × control). a** Beginning, **b** drought 1, **c** rewetting 1, **d** drought 2, **e** rewetting 2, and **f** the cumulative CO$_2$ fluxes, for the control (blue) and DRW (yellow) treatments on soil from either mono-specific or 3-species mixed stands. The most parsimonious model $R^2$ (marginal and conditional), standard error bars, and the significant differences between the control treatment, DRW treatment, mono-specific stands, and mixed stands at each experimental period, indicated by lower-case letters, are from two GLMMs run on individual CO$_2$ flux measurements and on cumulative CO$_2$ fluxes. The asterisk indicates a scale and unit change; calculated cumulative values are only rough estimates because our measurements did not cover the initial rewetting dynamics.

significant treatment × experimental period interaction (Table 1). However, in contrast to CO$_2$ fluxes, the negative drought effect was equally strong on N$_2$O fluxes in both cycles (−58% and −60% on average in the first and second cycle, respectively). In the first DRW cycle, soil rewetting also resulted in marginally significant higher N$_2$O fluxes compared to the control (+12%), but this increase was much lower than for CO$_2$ and not present in the second cycle (Table 1, Fig. 2c, e). Whether the soils were from mixed tree species or monospecific stands did not affect the N$_2$O fluxes nor their responses to repeated DRW (Table 1, Fig. 2). There were no apparent root trait legacy effects on N$_2$O fluxes, but soil parameters had an influence, with higher pH, C, and clay contents, and lower bulk density associated to higher N$_2$O fluxes (Table 1, Supplementary Fig. 2). The N$_2$O flux cumulated over the entire experiment was lower with repeated DRW compared to control conditions (−10%; Fig. 2f). Tree species number significantly changed the cumulative N$_2$O fluxes depending on the DRW treatment (significant species number × treatment interaction; Table 1, Fig. 2f), with lower cumulative N$_2$O fluxes from mono-specific stand soils (−21%) than from mixed stand soils after repeated DRW. The same soil parameters identified for N$_2$O flux dynamics also positively influenced the cumulative N$_2$O flux (Table 1).

**Microbial resistance and recovery**. CO$_2$ and N$_2$O flux resistance index values ranged from 0.01 to 2.1 unitless (average 0.46) and

from 0.0004 to 2.0 (average 0.41), respectively; zero indicates no resistance and gas flux cessation in the drought treatment, 1 indicates identical fluxes in control and drought treatment, and >1 indicates higher gas flux than the control average. Microbial CO$_2$ flux resistance to drought decreased between DRW cycles (−21% on average; Table 1, Fig. 3a). CO$_2$ flux in soil from mixed stands showed higher resistance compared to monospecific stands during both DRW cycles (+28% on average). In addition, we also observed a negative legacy effect of root trait functional dispersion (FDis), indicating that CO$_2$ flux resistance was higher when root functional traits were more similar. N$_2$O flux resistance to drought remained constant between the two cycles and was positively related to tree species number as observed for CO$_2$ fluxes (Table 1, Fig. 3b). CO$_2$ and N$_2$O flux recovery index values ranged from 0.19 to 5.1 unitless (average 1.47) and from 0.001 to 3.6 (average 1.23), respectively; 0 indicates no gas flux recovery, 1 indicates 100% recovery in relation to average control values, and >1 indicates higher gas flux than the control average. The CO$_2$ flux recovery decreased between DRW cycles (−6%), a decrease that appears to be primarily in soils from mixed species forests, but model results did not show neither a significant tree species number effect nor a tree species number × DRW interaction (Table 1, Fig. 3c). Finally, N$_2$O flux recovery marginally decreased between DRW cycles (−1%) and was partly explained by root trait legacy through root CWM, indicating a positive correlation with conservative root traits (larger diameter and higher tissue

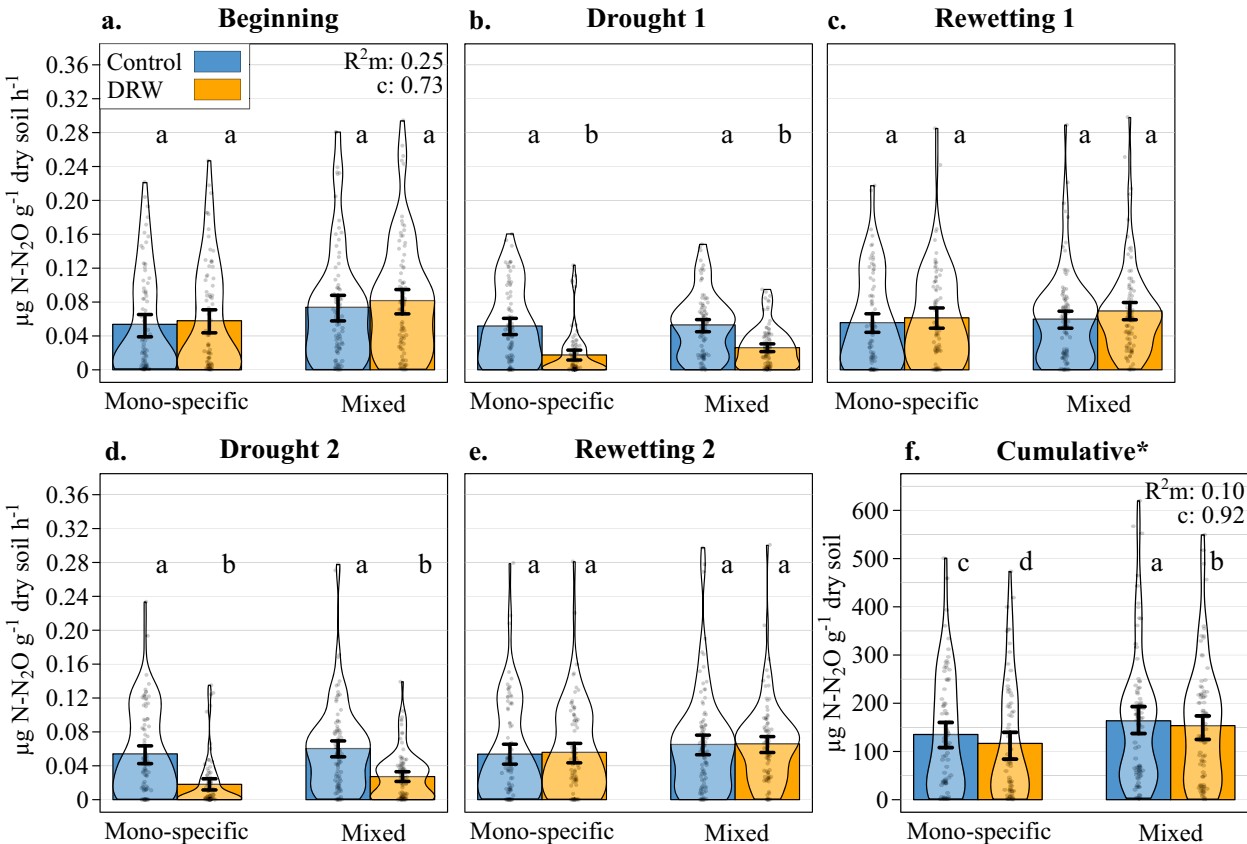

**Fig. 2 N$_2$O fluxes (µg N–N$_2$O × g$^{-1}$ dry soil × h$^{-1}$) over the five DRW experimental periods (microcosms: $n = 90$ monospecific × DRW, $n = 90$ monospecific × control, $n = 102$ mixed × DRW, $n = 102$ mixed × control).** **a** Beginning, **b** drought 1, **c** rewetting 1, **d** drought 2, **e** rewetting 2, and **f** the cumulative N$_2$O fluxes, for the control (blue) and DRW (yellow) treatments on soil from either monospecific or 3-species mixed stands. The most parsimonious model $R^2$ (marginal and conditional), standard error bars, and the significant differences between the control treatment, DRW treatment, monospecific stands, and mixed stands at each experimental period, indicated by lowercase letters, are from two GLMMs run on individual N$_2$O flux measurements and cumulative N$_2$O fluxes. The asterisk indicates a scale and unit change; calculated cumulative values are only rough estimates because our measurements did not cover the initial rewetting dynamics.

and length density), while tree species number had no effect on N$_2$O flux recovery (Table 1, Fig. 3d).

**Dissolved organic carbon and total nitrogen.** The amount of DOC extracted from soils at the end of the experiment was not affected by any of the treatment factors (Table 1, Fig. 4a). There was only a marginally significant soil parameter effect with lower DOC associated with higher soil pH, C and clay contents, and lower bulk density. Conversely, somewhat less TDN was measured in soils subjected to repeated DRW compared to the control (−6%), but there was neither a tree species number effect nor a tree species number × DRW interaction (Table 1, Fig. 4b). The same soil parameters that were associated with lower DOC were also associated to higher TDN values (Supplementary Fig. 2). Finally, we observed a root CWM effect on TDN, with higher amounts of TDN associated with more acquisitive root traits (Table 1, Supplementary Fig. 2).

**Metabolic quotient.** The metabolic quotient (qCO$_2$), measured as the ratio between the basal respiration rate and microbial biomass, is an estimation of microbial stress[32]. After two DRW cycles, basal respiration was higher in DRW treatment soils compared to control soils but was not affected by tree species mixing (Supplementary Table 4, Supplementary Fig. 4a). Microbial biomass was not affected by the DRW treatment but was overall marginally lower in mixed stands compared to

monospecific stands (Supplementary Table 4, Supplementary Fig. 4b). In turn, qCO$_2$ increased significantly compared to controls (+24% on average; Table 1, Fig. 4c). Although qCO$_2$ was higher in the DRW treatment for both monospecific and mixed stands (+34.7% and +13.6% respectively), the DRW effect was stronger (i.e., qCO$_2$ was higher) in monospecific stands (significant tree species number × DRW treatment interaction). Soil parameters and root CWM had an effect on qCO$_2$, indicating higher qCO$_2$ values with high bulk density and low pH, C, and clay contents as well as with acquisitive root traits (higher specific root length, N content, and ectomycorrhizal colonization intensity; Table 1, Supplementary Fig. 2).

**Discussion**
This study examined how drying-rewetting (DRW) cycles influence soil microbial activity related to C and N cycling in soils from different mature, natural European forests composed of one or three tree species. Despite a wide range of soil and forest types, including 13 tree species and 34 species combinations across four countries, our results showed a robust association between mixed tree species forests and higher resistance of soil microbial respiration (CO$_2$ fluxes) and denitrification (N$_2$O fluxes) as well as with lower soil microbial stress levels (qCO$_2$) in response to repeated DRW cycles, a scenario expected to become more common with ongoing climate change. However, this effect of mixed tree species forests did not extent to cumulative gas fluxes,

Table 1 The most parsimonious model results: $R^2$ marginal ($R^2$m), and $R^2$ conditional ($R^2$c), estimated slope (Est.), standard error (SE), degree of freedom (df), $t$-value, and $p$-values for the response variables $CO_2$ and $N_2O$ fluxes, cumulative $CO_2$ and $N_2O$ fluxes, $CO_2$ and $N_2O$ resistance and recovery indices, metabolic quotient (q$CO_2$), dissolved organic carbon (DOC), and total dissolved nitrogen (TDN). Red and blue estimate values indicate positive and negative relationships, respectively, and based on estimate values not significance. Explanatory variables are abbreviated as: DRW treatment (DRW), tree species number (Sp.No.), tree species number and DRW treatment interaction (Sp.No.:DRW), at the four DRW stages (drought 1, D1; rewetting 1, R1; drought 2, D2; rewetting 2, R2), the topsoil properties (Soil), absorptive root functional dispersion (Root FDis), absorptive root community weighted mean traits (Root CWM), the change between the first and second DRW cycle (Cycle), and tree species number and cycle interaction (Sp.No.:Cycle). Dashes indicate explanatory variables not retained in the most parsimonious model, $p$-values are coded as such: 0.1> and <0.05 "."; 0.05> and <0.01 "*"; 0.01> and <0.001 "**", 0.001>"***", variables were sometimes retained but not significant. The principle explanatory variables (DRW, D1, R1, D2, and R2) are not shown for the response variables accompanied by the dagger symbol "†", because the main effect outputs of the variables retained in interactions are not interpretable when higher level interactions are significant.

| | $CO_2$ flux† | | | | | | $N_2O$ flux† | | | | | |
| | $R^2$m = 0.329 | | $R^2$c = 0.467 | | AIC = 5082.1 | | $R^2$m = 0.252 | | $R^2$c = 0.73 | | AIC = 377.6 | |
| | Est. | SE | df | $t$-value | $p$-value | | Est. | SE | df | $t$-value | $p$-value | |
| Sp.No. | – | – | – | – | – | | – | – | – | – | – | |
| Sp.No.:DRW | – | – | – | – | – | | – | – | – | – | – | |
| D1:DRW | −1.18 | 0.13 | 1809.0 | −9.06 | < 2e−16 | *** | −0.33 | 0.04 | 1787.6 | −8.86 | < 2e−16 | *** |
| R1:DRW | 0.35 | 0.13 | 1809.0 | 2.67 | 0.01 | ** | 0.06 | 0.04 | 1787.6 | 1.72 | 0.09 | . |
| D2:DRW | −1.55 | 0.13 | 1809.0 | −11.94 | < 2e−16 | *** | −0.36 | 0.04 | 1787.6 | −9.57 | < 2e−16 | *** |
| R2:DRW | 0.39 | 0.13 | 1809.0 | 3.04 | 2.4E−03 | ** | 0.02 | 0.04 | 1787.6 | 0.43 | 0.67 | |
| Soil | – | – | – | – | – | | 0.13 | 0.03 | 64.3 | 5.06 | 3.7E−06 | *** |
| Root FDis | – | – | – | – | – | | – | – | – | – | – | |
| Root CWM | 0.15 | 0.04 | 6.2 | 3.47 | 0.01 | * | – | – | – | – | – | |

| | Cumulative $CO_2$ flux | | | | | | Cumulative $N_2O$ flux | | | | | |
| | $R^2$m = 0.303 | | $R^2$c = 0.756 | | AIC = 421.8 | | $R^2$m = 0.103 | | $R^2$c = 0.915 | | AIC = 444.4 | |
| | Est. | SE | df | $t$-value | $p$-value | | Est. | SE | df | $t$-value | $p$-value | |
| DRW | −0.25 | 0.06 | 5.4 | −4.21 | 7.2E−03 | ** | −0.27 | 0.06 | 14.8 | −4.37 | 5.6E−04 | *** |
| Sp.No. | – | – | – | – | – | | 0.04 | 0.17 | 61.8 | 0.22 | 0.83 | |
| Sp.No.:DRW | – | – | – | – | – | | 0.20 | 0.08 | 269.7 | 2.60 | 0.01 | ** |
| Soil | – | – | – | – | – | | 0.24 | 0.10 | 59.7 | 2.51 | 1.5E−02 | * |
| Root FDis | – | – | – | – | – | | – | – | – | – | – | |
| Root CWM | 0.20 | 0.03 | 63.3 | 5.88 | 1.7E−07 | *** | – | – | – | – | – | |

| | $CO_2$ flux resistance | | | | | | $N_2O$ flux resistance | | | | | |
| | $R^2$m = 0.187 | | $R^2$c = 0.369 | | AIC = 990 | | $R^2$m = 0.07 | | $R^2$c = 0.781 | | AIC = 354.9 | |
| | Est. | SE | df | $t$-value | $p$-value | | Est. | SE | df | $t$-value | $p$-value | |
| Sp.No. | 1.86 | 0.65 | 59.2 | 2.88 | 5.54E−03 | ** | 0.24 | 0.10 | 59.1 | 2.42 | 0.02 | * |
| Cycle | −0.30 | 0.10 | 306.8 | −3.03 | 2.7E−03 | ** | – | – | – | – | – | |
| Sp.No.:Cycle | – | – | – | – | – | | – | – | – | – | – | |
| Soil | – | – | – | – | – | | −0.08 | 0.05 | 61.5 | −1.52 | 0.13 | |
| Root FDis | −0.89 | 0.38 | 59.2 | −2.35 | 0.02 | * | – | – | – | – | – | |
| Root CWM | – | – | – | – | – | | – | – | – | – | – | |

| | $CO_2$ flux recovery | | | | | | $N_2O$ flux recovery | | | | | |
| | $R^2$m = 0.011 | | $R^2$c = 0.176 | | AIC = 1011.5 | | $R^2$m = 0.08 | | $R^2$c = 0.518 | | AIC = 489.7 | |
| | Est. | SE | df | $t$-value | $p$-value | | Est. | SE | df | $t$-value | $p$-value | |
| Sp.No. | – | – | – | – | – | | – | – | – | – | – | |
| Cycle | −0.21 | 0.09 | 305.8 | −2.22 | 0.03 | * | −0.09 | 0.05 | 290.0 | −1.93 | 0.06 | . |
| Mix:Cycle | – | – | – | – | – | | – | – | – | – | – | |
| Soil | – | – | – | – | – | | – | – | – | – | – | |
| Root FDis | – | – | – | – | – | | – | – | – | – | – | |
| Root CWM | – | – | – | – | – | | 0.09 | 0.03 | 59.7 | 2.96 | 4.5E−03 | ** |

| | DOC | | | | | | TDN | | | | | |
| | $R^2$m = 0.054 | | $R^2$c = 0.556 | | AIC = 1327.4 | | $R^2$m = 0.433 | | $R^2$c = 0.754 | | AIC = 2332.8 | |
| | Est. | SE | df | $t$-value | $p$-value | | Est. | SE | df | $t$-value | $p$-value | |
| DRW | – | – | – | – | – | | −0.99 | 0.45 | 317.2 | −2.22 | 0.03 | * |
| Sp.No. | – | – | – | – | – | | – | – | – | – | – | |
| Sp.No.:DRW | – | – | – | – | – | | – | – | – | – | – | |
| Soil | −0.26 | 0.14 | 33.28 | −1.86 | 0.07 | . | 1.19 | 0.39 | 63.8 | 3.06 | 3.3E−03 | ** |
| Root FDis | – | – | – | – | – | | – | – | – | – | – | |
| Root CWM | – | – | – | – | – | | −0.87 | 0.35 | 64.06 | −2.46 | 0.02 | * |

| | q$CO_2$ | | | | | |
| | $R^2$m = 0.187 | | $R^2$c = 0.368 | | AIC = 989.8 | |
| | Est. | SE | df | $t$-value | $p$-value | |
| DRW | 0.65 | 0.19 | 11.3 | 3.38 | 0.01 | ** |
| Sp.No. | 0.26 | 0.18 | 105.13 | 1.47 | 0.14 | |
| Sp.No.:DRW | −0.45 | 0.20 | 278.74 | −2.26 | 0.02 | * |
| Soil | −0.16 | 0.06 | 32.5 | −2.52 | 1.7E−02 | * |
| Root FDis | – | – | – | – | – | |
| Root CWM | −0.15 | 0.07 | 13.3 | −2.18 | 4.8E−02 | * |

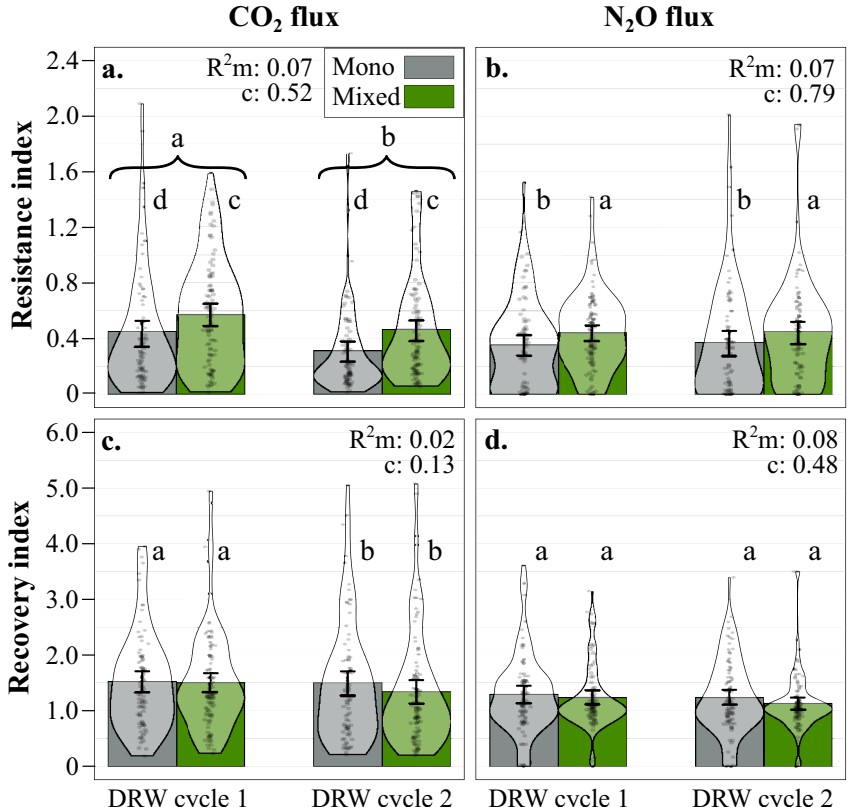

**Fig. 3 Aerobic respiration and denitrification resistance and recovery indices. a** $CO_2$ flux resistance, **b** $N_2O$ flux resistance, **c** $CO_2$ flux recovery, and **d** $N_2O$ flux recovery indices for soils from mono-specific stands (Mono, gray; $n = 90$ index values for each gas) or 3-species mixed stands (Mixed, green; $n = 102$ index values for each gas) for each DRW cycle (DRW cycle 1 and DRW cycle 2). The GLMM most parsimonious model results: $R^2$ marginal (m) and conditional (c) and significant differences, indicated by lowercase letters, and standard error bars.

microbial recovery, or potential dissolved carbon or nitrogen leaching (DOC and TDN).

Consistent with previous findings[7,33] and our hypothesis, DRW events had a strong effect on microbial activities with a sharp decline following drought periods for both aerobic respiration and denitrification that decreased more during the second cycle for aerobic respiration. This decline is likely due to a direct drought effect on microbial physiology (reduction of cellular water potential) and on organic substrate diffusion, and oxygen diffusion in the case of aerobic respiration, that causes a decline in enzymatic activity[7]. Aerobic respiration seven days following rewetting was much higher than control values, but this was less pronounced in the second cycle. This, along with the stronger aerobic respiration decline following the second drought, could indicate the microbial mortality and/or a shift in the microbial community composition or physiological strategies[8] to stress-avoidance through declined activity during stress[9]. Since denitrification rates returned to pre-stress levels after rewetting, there was no indication of a loss in the denitrification taxa group during drought, instead the microorganisms appear tolerant to very low soil water potential through decreased activity or accumulation of protective molecules[7].

Lower cumulative aerobic respiration and denitrification for the DRW compared to the control treatment may suggest that the decrease in microbial activity during drought periods was not compensated by the increase in fluxes following the rewetting periods, which supports previous findings[34,35]. The lower cumulative aerobic respiration and denitrification were potentially due to drought length, which impacts how much microbial

processes are reduced[36], and potentially exacerbated by the rewetting period length[37], which was relatively short in our study (Supplementary Fig. 3). Moreover, we did not measure the pulse of aerobic respiration (Birch effect) and denitrification activity immediately after rewetting[38,39], which could account for a substantial portion of net gas fluxes during DRW cycles. Indeed, the complexity of our experiment and the large number of soil microcosms included made it impossible to track flux changes during drying and rewetting with more frequent or continuous measurements, thus requiring cautious interpretation of the cumulative flux data. Furthermore, shifts in soil microbial gas fluxes depend on numerous variables (e.g., drought severity, rewetted soil water content, soil texture and compaction, substrate accumulation, as well as local temperatures)[40], and microbial responses could shift with successive DRW cycles[7]. Although cumulative $CO_2$ fluxes were not affected by tree species number, they were affected by root functional traits, specifically the community weighted mean (CWM) traits of the absorptive roots (i.e., the lowest three orders of roots) with higher cumulative $CO_2$ fluxes associated with conservative root traits (i.e., roots with larger diameter and higher tissue and length density). Our results showed that cumulative denitrification was positively influenced by mixed stands during DRW; the denitrification rate was closer, though still lower, to the control rate in the soil from the mixed stands. Yet, we are unable to extrapolate greenhouse gas $N_2O$ fluxes from this cumulated value since $N_2$ production was not measured, the denitrification $N_2O:N_2$ ratio is dependent on soil water content[41], and the caveats associated with the cumulative values mentioned above.

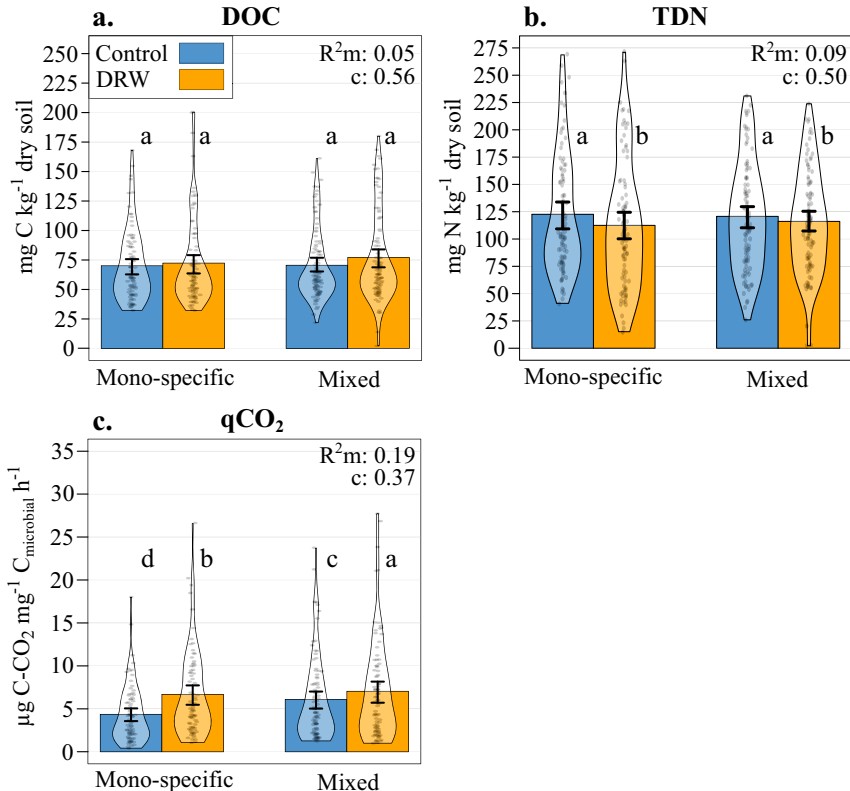

**Fig. 4 Microbial stress and dissolved carbon and nitrogen following two DRW cycles. a** total dissolved organic carbon (DOC; mg C kg$^{-1}$ dry soil), **b** total dissolved nitrogen (TDN; mg N kg$^{-1}$ dry soil), and **c** metabolic quotient (µg C–$CO_2$ mg$^{-1}$ microbial C h$^{-1}$) results for control (blue) and DRW (yellow) treatments on soil from monospecific or 3-species mixed stands (microcosms: $n = 90$ monospecific × DRW, $n = 90$ monospecific × control, $n = 102$ mixed × DRW, $n = 102$ mixed × control). The GLMM most parsimonious model results: $R^2$ marginal (m) and conditional (c) and significant differences, indicated by lowercase letters and standard error bars.

The absence of a tree mixture effect on microbial respiration is in contrast to a recent meta-analysis reporting an overall positive relationship between plant diversity and microbial respiration[30]. However, we only compared three species mixtures to single species forests, while the meta-analysis included a larger gradient in species numbers in multiple ecosystem types and reported stronger effects with increasing species numbers. On the other hand, to our knowledge, our study is among the first to address this question for well-established (>30 years old) natural forest communities. A study on a 7-year old planted experimental forest in Belgium with a one to four tree species richness gradient also did not find a significant tree mixing effect on microbial respiration in response to DRW[42], while another study in the same forest found that species mixing could influence microbial community composition through changes in fungal-to-bacterial growth[43].

Metabolic quotient (qCO$_2$) is an indicator reflecting the energy requirement for cell maintenance and microbial carbon use, i.e., cell activity and resources dedicated to cell maintenance instead of cell synthesis[32]. Higher qCO$_2$ in the DRW treatment suggests higher microbial stress compared to control treatments, meaning more energy is spent on microbial maintenance (respiration) than growth (biomass), and thus potentially indicating the poor energy-use efficiency[7]. Higher stress with DRW could lead to higher microbial mortality and microbial necromass resulting in higher C and nutrient loss through leaching over the long term. After two DRW cycles, microbial biomass was not negatively affected by DRW (Supplementary Fig. 4), and there was only a nonsignificant trend for higher potential DOC leaching in the

DRW treatment, while potential TDN leaching was lower compared to the control treatment (Fig. 4). The smaller difference in qCO$_2$ between DRW and control conditions observed in mixed stands compared to monospecific stands indicates that the microbial community was less affected or stressed by DRW and more energy efficient in mixed species stands. This is potentially due to higher nutrient diversity or availability associated with higher tree diversity[19]. A meta-analysis found qCO$_2$ to be correlated with soil organic C content[44]. Although differences in soil C concentrations between monospecific and mixed plots in our study were minimal (Supplementary Table 2), the quality of organic C could have played a role.

Dissolved organic carbon (DOC) and total dissolved nitrogen (TDN) are important in C and N sources for soil microorganisms[45], but could be leached from the system leading to reduced C and N availability for soil microorganisms. Less TDN in soils subjected to DRW cycles may indicate higher microbial assimilation into biomass or denitrification[39], but overall suggests a faster exhaustion of soil mineral nitrogen with DRW. If this pattern remains constant with increased drought frequency and severity, future soils subjected to DRW cycles may become more N-limited. This also may be one cause of the increased microbial stress (qCO$_2$) seen with DRW. In the literature, the effects of DRW on DOC and TDN soil concentrations and their potential leaching from the ecosystem are not clear. Previous studies have found an increase in DOC and TDN with DRW cycles[15,46], no effect[8,46], or a short-term change but no net effect[47,48]. This discrepancy appears linked to soil type, habitat, and/or drought frequency[15,46] and intensity[44].

As hypothesized, mixed tree species mitigated negative DRW effects, i.e., both microbial respiration and denitrification decreased less in soils from mixed compared to monospecific stands in response to drought. Higher substrate quantity, quality, or diversity provided by higher tree species number[19,22,23,26] could lead to higher microbial diversity and more drought-resistant taxa[29]. In addition, more diverse substrates could facilitate the production of protective molecules[9] or augment microbial efficiency[49], thus allowing the microbial community to maintain higher activity during drought. The lack of tree species number × cycle interaction for aerobic respiration resistance and the decrease in resistance after the second cycle indicate that, while higher tree species number increases drought resistance, it does not avoid the decrease in resistance in response to repeated stress. The observed tree mixture effects may also be associated to a difference in microbial community composition, for example a higher fungi:bacteria ratio—fungi typically being more drought resistant[14,50]—but this may rather be linked to forest composition and not species richness per se[51]. Regardless of the specific microbial group, the selection of soil microorganisms more resistant to repeated drought was likely a reason for lower stress levels, indicated by $qCO_2$ values, in soils from mixed stands.

Although aerobic respiration resistance was higher in mixed stands, it also decreased with increasing root trait functional dispersion (Table 1). This not only underlines a disconnect between tree taxonomic diversity and functional diversity but also could suggest that the importance of tree species number may outweigh tree functional diversity when considering multiple forest types, which is contrary to findings on single forest types[30,31]. We considered the functional dispersion of morphological and chemical tree absorptive root traits, which do not necessarily represent all aspects of tree functional diversity. More functionally diverse root traits should lead to increased belowground niche partitioning and better exploitation of soil resources[28], which may reduce resource availability for soil microorganisms. Higher root functional diversity may also dilute optimum soil resource concentrations (resource concentration hypothesis) leading to decreased microbial efficiency[52]. Both mechanisms could explain why higher functional root diversity may diminish microbial resistance to recurrent drought.

Microbial recovery was not affected by tree species number for neither aerobic respiration nor denitrification, and we did not observe any correlation between aerobic respiration resistance and recovery indices, suggesting no trade-off between growth and stress survival (Supplementary Table 3)[31,53]. Denitrification resistance and recovery indices, on the other hand, were significantly correlated but positively, which does not demonstrate a trade-off either (Supplementary Table 3). These results suggest that ecosystem functioning may be less affected by drought in more tree species diverse forests due to microbial communities that are able to sustain more stable biogeochemical cycling rates during the drought, but not necessarily after rewetting. However, it is not known how results may have varied if the experiment had been conducted in situ, i.e., with the vegetation present, even though it would be difficult to separate microbial heterotrophic respiration from autotrophic respiration.

In conclusion, our results show that forest soil microbial driven C and N processes will likely be influenced by the predicted increase in drought events, but tree species diversity could help microbial communities resist better by mitigating negative drought effects. Microbial communities in more diverse forests will be better able to remain active and continue C and N cycling thus potentially better sustaining ecosystem functioning and stability compared to monospecific forests. Remarkably, these positive tree mixture effects on microbial resistance under drought were robust across very different forest types and distinct soil types over a large geographical distribution. These results may help to predict the resistance of forest soil microbial communities to DRW as well as how these ecosystems will be influenced by future climatic changes.

## Methods

**Forest sites and soil sampling.** The sites used in this study are part of a permanent network of existing mature forest plots across Europe established in 2011–2012 (see Baeten et al.[54] for detailed descriptions). We included four sites ranging over a large climatic gradient: North Karelia (Finland), Białowieża (Poland), Râșca (Romania), and Colline Metallifere (Italy), which correspond to typical boreal forests, hemiboreal mixed broadleaved-coniferous, montane mixed beech, and Mediterranean thermophilous, respectively (Supplementary Table 1, Supplementary Fig. 1). At each site, we selected 30 m × 30 m forest plots dominated by either one tree species (monospecific stands) or by three co-dominating tree species, hereafter referred to as mixed stands, resulting in a total of 34 species combinations (species were considered co-dominant if they composed >15% of the stand; see Supplementary Data file 1 for plot and tree species information). Each site differed in total species numbers, species identity, and species combinations (Supplementary Table 1). There were two replicates per tree species for the monospecific plots of each site, except for *Picea abies* and *Quercus robur*, which were only replicated once and *Betula pendula* which had no mono-specific plot in Białowieża. There was a minimum of three mixed species plot replicates per site that were composed of any of the target species present at the site (Supplementary Table 1), i.e., the replicate mixed plots at each site did not necessarily have the same tree species combinations. There were 64 plots in total. The sampling design with the total plot number, their distribution over four forest ecosystems, and including a wide range of tree species is well suited to address the generality of our hypothesis that microbial responses to DRW cycles are modified by tree species mixing but poorly suited to identify site-specific patterns with plot numbers too limiting within specific sites for robust testing.

Within each plot, we selected five tree triplets, a triplet being a triangle of three tree individuals within a maximum distance of 8 m from each other and no obstructing tree individuals within the triangle. Each triplet was composed of either the same species in the monospecific stands (monospecific triplet) or the three tree species present in the mixed stands (mixed triplet). At the estimated tree individual size weighted (based on individual diameter at breast height) center within the triangle, we collected five soil cores from the topsoil (10 cm deep, 5.3 cm diameter) after the litter layer had been removed. The five soil cores were spaced at roughly 35 cm from each other circling the center point (approximate sampled area 50 cm × 50 cm). A depth of 10 cm was selected because it is the standard topsoil sampling depth in soil ecology, has the highest soil microbial activity, and is under the most influence from the plant community[19]. All soil cores from each sampling location (i.e., tree triplet) within a plot were then sieved together through a 2 mm sieve and air-dried immediately after sampling for transportation and experiment preparation.

**Experimental design.** The soils collected from the 64 forest plots at the four sites were split into six replicate microcosms, yielding a total of 384 microcosms that were housed at the Montpellier European Ecotron CNRS in Montpellier, France. Each microcosm contained 95 g dry weight of soil in a glass vial (soil volume 51–72 ml; air volume 259–279 ml), initially incubated at 80% of water holding capacity (WHC) using deionized water, 25 °C, no light, and 40% relative air humidity (the vials were covered with Parafilm® to allow gas exchange but to prevent soil desiccation) for 3 weeks to reactivate the microbial community (Supplementary Fig. 3). After this acclimation period, half of the microcosms (192, i.e., $n = 3$ per plot) was assigned to a drying-rewetting (DRW) treatment and the other half (192, i.e., $n = 3$ per plot) to a control treatment. Maximum microbial mineralization activity appears to be reached between 60% and 80% WHC[55]. We chose 80% to ensure soils were entirely and homogeneously humid; very sandy soils with a low WHC, such as those from the Polish site, were not completely wetted at the typically chosen 60% WHC. Each treatment replicate was housed in a 2 m³ individual growth chamber ($n = 6$). Within each chamber, the microcosms were randomly distributed on a single shelf and re-randomized weekly. The DRW treatment was defined as two DRW cycles while the soils in the control treatment were maintained at 80% WHC throughout the experiment (Supplementary Fig. 3). Water content was adjusted gravimetrically 2–3 times a week.

Due to the large latitudinal distribution and varying soil and climate conditions of the sites (Supplementary Table 1), the soil microbial communities do not necessarily have the same degree of drought history and adaptation[56]. We therefore applied a site-specific drought treatment representative of each of the four study sites, i.e., site-specific drought intensity and duration. We used the permanent wilting point as a water stress threshold indicator since there is not a known microbial equivalent. The permanent wilting point was measured using a pressure plate extractor (1500F2, Soilmoisture Equipment Corp., Santa Barbara, USA) at pF 4.2 (15.5 bar) for the plots with the fastest and slowest drying soils of each site. The soil drying speed, i.e., the number of days it took for the soil to dry from 80% WHC down to constant weight, was measured gravimetrically for each plot using a subsample of soil that was subsequently excluded from the experiment. We then averaged the permanent wilting

point values per site and designated this average as the drought intensity: Colline Metallifere 11% $H_2O$ $g^{-1}$ dry soil, Râşca 30%, Białowieża 12%, and North Karelia 12%. The beginning of the drought was considered the moment the soil water content arrived at this threshold. The drought duration was calculated using the forest drought history data from Grossiord et al.[56] as the average annual number of days the relative extractable water (REW) dropped below 0.2 (unitless) over the 1997–2010 period. REW is the ratio of available soil water to maximum extractable water (i.e., WHC), ranging between the field capacity (REW 1.0) and the permanent wilting point (REW 0.0)[56]. Plants are in non-water limited conditions when REW is >0.4 and water limited when REW is <0.4; we therefore chose a REW threshold of 0.2 to ensure both plant and soil microorganisms were in water-stressed conditions. Drought duration was therefore 38 days for Colline Metallifere, 12 days for Râşca, 8 days for Białowieża, and 0 days for North Karelia per DRW cycle; total experimental drought duration over the two cycles was thus twice that occurring under natural field conditions. Although North Karelia soils were never subjected to drought (as defined above) over the 14 years of reference (drought duration: 0 days), we considered the drying period from 80% WHC to the drought intensity threshold (lasting from 7 to 12 days; Supplementary Fig. 3) as an already considerable stress for the microbial community, and the soils were also kept at the drought intensity threshold during the gas flux measurement period (2 days). The microcosms from the remaining three sites were kept at the drought intensity threshold for the site-specific drought duration plus the two days of gas flux measurements. All other environmental conditions simulated in the growth chambers (temperature, humidity, and darkness) remained unchanged for all microcosms during the duration of the experiment.

As a consequence of the site-specific drought duration, as well as different soil drying speeds, the DRW treatments differed in length. In order to ensure the microbial communities were active for the same duration, which better allows analysis of microbial activity from different sites, we staggered the beginning of the first drying-rewetting cycle so that all microcosms finished the two DRW cycle treatment at approximately the same time. We refer to the period between the first gas measurement and the beginning of the first drying period as the buffer period (Supplementary Fig. 3). Although the buffer period could have led to an exhaustion of some of the readily available resources for microorganisms, comparing the activity of the DRW treatment microcosms to parallel control microcosms following exactly the same dynamics of potential resource exhaustion should have avoided confounding of drought responses with different levels of resource exhaustion. Since it was unfeasible to stagger each microcosm individually, we regrouped the microcosms by site and by soil drying speed (i.e., drying period duration) into eight groups (two groups of plots with more slowly and rapidly drying soils, respectively for each site, Supplementary Fig. 3).

## $CO_2$ and $N_2O$ flux measurements

Soil microbial respiration and denitrification activities were estimated by measuring $CO_2$ and $N_2O$ fluxes in the microcosms. We measured the $CO_2$ and $N_2O$ fluxes at the end of the 3-week acclimation period (Supplementary Fig. 3, m0), at the end of the first and second drought periods (Supplementary Fig. 3, m1 and m3), and 7 days following the rewetting of the drought exposed microcosms to 80% WHC (Supplementary Fig. 3, m2 and m4). We used this seven day delay to avoid the Birch effect[9,38]. The second DRW cycle started immediately after the post-rewetting flux measurement. Flux measurements were done concurrently in both the DRW and control microcosms of the same group.

For the $CO_2$ measurements, we sealed the microcosms for 23 h to allow $CO_2$ accumulation, which we then measured using a MicroGC (S-Series, SRA Intruments, Marcy l'Etoile, France). We subsequently replaced the air in the microcosms with 90% helium and 10% acetylene to prevent $N_2O$ reduction to $N_2$ (allowing the measurement of denitrified N as $N_2O$), then incubated the microcosms for another 23-h period to again allow gas accumulation under anaerobic condition. The $N_2O$ concentration was then measured using a GC CP-3800 equipped with an electron capture detector (Varian, Palo Alto, USA). Flux rates were calculated as the amount in µg of $CO_2$ or $N_2O$ produced per gram of soil per hour (µg $C–CO_2$ or $N–N_2O$ $h^{-1}$ $g^{-1}$ dry soil).

## Cumulative $CO_2$ and $N_2O$ flux and resistance and recovery index calculations

The cumulative $CO_2$ and $N_2O$ fluxes were estimated over the entire experimental period: we multiplied the $CO_2$ and $N_2O$ flux rates, estimates of gas flux at a single point in time, by the duration of the experimental period preceding that measurement (Supplementary Fig. 3; the post-drought measurement by the drought period duration, and the post-rewetting measurement by the rewetting duration). The beginning flux measurement was multiplied by the buffer period (Supplementary Fig. 3, m0). Since we did not measure gas fluxes directly after the drying periods (when the soils dried from 80% WHC down to the drought intensity threshold before the start of the drought period), we averaged the gas flux measurement before the drying period and the measurement at the end of the drought period and then multiplied it by the drying period duration. We then summed these period cumulative flux estimates yielding an estimate of the total $CO_2$ and $N_2O$ fluxes during the entire experiment. It is important to note that these are only rough estimates based on relatively few measurements and ignoring potentially important fluxes during the Birch effect period. These data should be interpreted cautiously.

We followed Nimmo et al.[53] for the calculations of indices of resistance and recovery of microbial activity. Specifically, resistance was calculated by dividing the values of $CO_2$ and $N_2O$ flux rates measured in the DRW treatment by those measured in the control treatment at the end of each drought period (Supplementary Fig. 3, m1 and m3). Recovery was calculated by dividing the values of $CO_2$ and $N_2O$ flux rates measured in the DRW treatment by those measured in the control treatment at the end of each rewetting period (Supplementary Fig. 3, m2 and m4). Since there were no defined control and DRW treatment microcosm replicate pairs, we divided the results of each individual DRW microcosm replicate by the average of the three control treatment results.

## Post-experiment soil analyses

After the end of experiment, the soil microbial biomass was estimated using the substrate-induced-respiration (SIR) method[57,58], which was then used to calculate the metabolic quotient ($qCO_2$) defined as the $C–CO_2$ respired per unit of microbial biomass (ng $C–CO_2$ $µg^{-1}$ $C_{mic}$ $h^{-1}$)[32]. The microbial biomass was estimated following Anderson and Domsch[58]: SIR rate (µl $C–CO_2$ $g^{-1}$ dry soil $h^{-1}$) × 40.04 + 0.37. The final $CO_2$ measurement of the experiment (Supplementary Fig. 3, m4) was considered a measure of the basal respiration rate. We then divided the basal respiration rate by the estimated microbial biomass to obtain $qCO_2$[32] and converted it to be expressed in ng $C–CO_2$ $µg^{-1}$ $C_{mic}$ $h^{-1}$.

The potentially leachable C and N were estimated by quantifying soluble organic C (DOC) and total soluble N (TDN) in the soils at the end of the experiment. DOC and TDN were measured using a method adapted from Jones and Willett[59] with a TOC analyzer equipped with a supplementary module for N (CSH E200V, Shimadzu, Kyoto, Japan). For the extraction, 30 ml of a 0.025 M $K_2SO_4$ solution were added to the soil at 70% WHC (wet soil weight equivalent to 10 g dry soil) with five glass balls and agitated for 30 min at 250 rpm at room temperature. The mixture was then centrifuged for 5 min at 2500 rpm at 4 °C, and the supernatant was passed through a 0.45 µm filter, which was then analyzed for DOC and TDN (mg C or N $kg^{-1}$ dry soil).

See Supplementary Data file 1 for mean and standard deviation values for all response variables (i.e., $CO_2$ and $N_2O$ fluxes, cumulative fluxes, DOC, TDN, $qCO_2$, microbial biomass, and resistance and recovery indices) at site, tree species number, and treatment levels.

## Statistics and reproducibility

The R software (version 3.5.3[60]) was used for all statistical analyses and figures; figures were made using the "pirateplot" function from the YaRrr! Package (version 0.1.5[61]), and the function "fviz_pca_biplot" from the factoextra package (version 1.0.6[62]), and the ggplot2 package (version 3.2.0[63]). The map of sampling locations (Supplementary Fig. 1) was created using the QGIS software (version 3.12.3).

Since tree functional characteristics can play a large role in microbial community composition and functioning[17,19,23,31], we also calculated the functional dispersion (FDis[64]) and the community weighted mean (CWM[65]) of absorptive root traits (absorptive roots defined as the first three most distal root orders[66]) measured from tree roots harvested from one of the soil cores we took for soil sampling for each individual plot. Soil core samples were kept frozen before roots were separated from mineral particles and other organic matter over a sieve cascade with tap water. Roots were separated by diameter into coarse (>2 mm diameter) and fine roots (≤2 mm in diameter). Fine roots were further separated into tree and understory roots. Tree fine roots were further divided into dead (which are hollow, brittle, and dark-colored) and live fine roots, which were then sorted by species (based on distinct color, architecture, morphology, and mycorrhizal types) and subsequently further divided based on their functions into absorptive and transport roots[66]. Ectomycorrhizal root tips were counted on absorptive tree roots using a binocular. All absorptive tree fine-root samples were scanned with a flat-bed scanner (resolution of 800 dpi) and scans were then analyzed using WinRhizo (Regent Instruments, Quebec, Canada, 2009) to quantify root length, surface area, volume, and diameter. Coarse root samples were also scanned to obtain coarse root volume, which was used together with the stone mass to calculate fine-earth volume ($cm^{-3}$) of each soil sample. Root samples were dried (minimum 72 h, 40 °C) and weighed. Carbon and nitrogen concentrations of milled absorptive fine-root samples were measured for samples pooled at the plot level using dry combustion (Elementar Vario El Cube). Absorptive root analysis results are provided in Supplementary Table 2. We chose absorptive root traits rather than the commonly used leaf traits to characterize functional trait characteristics of tree communities, because the majority of soil microorganisms are intimately associated with the rhizosphere and thus root traits[67]. CWM is a measure of the relative species abundance weighted trait values. FDis is a measure of the abundance weighted mean distance between the "trait space' of individual species. Both indices were calculated with the same standard root chemical and morphological traits (Supplementary Table 2) using the R function 'dbFD' in the FD package (version 1.0-12[68]). Due to difficulties in differentiating between *Quercus* species in root samples from some of the Italian plots, we were unable to determine mean absorptive root trait values at the plot level. We therefore used mean root trait values at the site level calculated from the mono-specific stands. Although the root trait values were not at the plot level, we were still able to determine the CWM and FDis indices at plot level since the root traits values were reported to tree species relative abundance in each plot. The relative abundance of

each tree species was calculated using the basal areas of the tree individuals used in the five plot tree triplets (three tree individuals per tree triplet). Within each plot, the basal areas of a tree species (including five or fifteen tree individuals depending on whether the plot was mixed or mono-specific, respectively) were summed and then reported to the total basal area of the 15 tree individual, giving the relative basal area of each tree species within each plot. In order to synthesize this data, we incorporated them in a principal component analysis (PCA) and extracted the first axis scores (explaining 52.8% of the variance; Supplementary Fig. 2). Although the evidence supporting a universal root economics spectrum (RES) for woody species is inconsistent[69–71], we consider our PCA1 axis as an acquisitive to conservative trait gradient with lower scores represented acquisitive root traits (high N content, specific root length, and ectomycorrhizal colonization intensity) and higher scores represented conservative traits (large diameter and high tissue density). The FDis was calculated following Laliberté & Legendre[64] based on all traits at the plot level. The mono-specific stands had a FDis value of zero, which limits FDis variability for half of the plots. Accordingly, there was one single FDis value per plot that was used in our statistical analyses.

Since soil microbial resistance and recovery are tied to soil parameters and resource availability[17,18], we also included major topsoil parameters (0–10 cm) known to affect microbial activity and/or community composition (Supplementary Table 2) measured previously during the FunDivEurope project[54] at the plot level. Similar to the CWM absorptive root traits, we incorporated the topsoil variables into a PCA using the function 'prcomp' from the factoextra package (version 1.0.6[62]) and extracted the first axis scores (explaining 52.5% of the variance; Supplementary Fig. 2) for a synthetic soil parameter measure for each individual plot. High PC1 scores are associated with higher pH, carbon content, and clay content and lower bulk density, the inverse is correlated with low PC1 scores.

We used generalized mixed-effects linear models (two-sided) using the lme4 package (version 1.1–21[72]) to assess the effects of the DRW treatment and the influence of the tree species number on microbial C and N-related parameters. The root FDis, root CWM PC1, and soil PC1 variables were included with the treatment × tree species interaction as explanatory variables. For the response variables (instantaneous $CO_2$ and $N_2O$ fluxes measured five times over the experiment and cumulative fluxes, DOC, and TDN leaching, qCO₂, and resistance and recovery indices), extreme values were removed (±3 times the IQR of all values for each variable). The soil collection site and plot as well as the growth chambers used for the incubation were included as random variables with plot nested within site. We did not include any climatic variables from the different sites, because they were highly correlated to site, which was already a random effect in the model. The model structure was as follows: response variable ~ Root FDis + Root CWM PC1 + Soil PCA axis + Treatment * Tree species number * Flux measurement time + (1|Chamber) + (1 | Site/Plot). The "Flux measurement time" variable, which identifies the times the five flux measurements were taken (i.e., beginning, drought 1, rewetting 1, drought 2, rewetting 2; Supplementary Fig. 3), was used only in the models that looked at the temporal dynamics of $CO_2$ and $N_2O$ fluxes. For the analysis of resistance and recovery indices, we did not keep the "Treatment" variable in the model since these indices were calculated using both the DRW and control treatment results (see above). Additionally, for the resistance and recovery indices, instead of a "Flux measurement time" variable, a "Cycle" variable was included to distinguish the microbial activity resistance and recovery of the first and second cycles; the "Cycle" result indicates the change between the first and second cycle. Model residuals were plotted to test for normality, and data was transformed (log2 or BoxCox) when normality was not met. We also verified for data homogeneity and model probability (Q–Q plots). In order to identify the most parsimonious model we used the R software (version 3.5.3) and the "dredge" function in the MuMIn package (version 1.43.6[73]) which uses the lowest Akaike information criteria (AIC) to rank all possible models with all possible combinations of the explanatory variables in the full model.

The data presented here is tied to specific spatial and temporal ecological conditions (e.g., forest drought history, tree species presence, microbial community composition, and soil property heterogeneity) which are susceptible to change. This makes exact study replication challenging and underlines the importance of including a wide range of conditions (e.g., multiple forest types, tree species, tree species combinations, climatic conditions, and soil types) as done here in order to explore general, potentially reproducible, trends oppose to site-specific trends.

**Reporting summary**. Further information on research design is available in the Nature Research Reporting Summary linked to this article.

## Data availability
The datasets generated during and/or analyzed during the current study are available from the corresponding author on reasonable request. This data is stored as excel files on a data portal associated with the FunDivEUROPE and SoilForEUROPE projects.

## Code availability
The codes created to analyze the datasets during the current study are available from the corresponding author on reasonable request.

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

## Acknowledgements

This research was part of the SoilForEUROPE project funded through the 2015–2016 BiodivERsA COFUND call for research proposals, with the national funders Agence Nationale de la Recherche (ANR, France), Belgian Science Policy Office (BELSPO, Belgium), Deutsche Forschungsgemeinschaft (DFG, Germany), Research Foundation Flanders (FWO, Belgium), and The Swedish Research Council (FORMAS, Sweden). We thank Janna Wambsganß and the team at the University of Freiburg for the root trait data, Bart Muys and Karen Vancampenhout at KU Leuven University for the soil texture data, the site managers Leena Finér, Bogdan Jaroszewicz, Olivier Bouriaud, and Filippo Bussotti, as well as Jakub Zaremba, Ewa Chećko, Iulian Dănilă, Timo Domisch and the SoilForEUROPE consortium for their assistance with the soil sampling, Jean Delariviere and Nancy Rakotondrazafy for their assistance with the DOC and TDN measurements conducted at the Eco&Sols BioSolTrop lab (LabEx Center Méditerranéen de l'Environnement et de la Biodiversité), Montpellier, France, and the Ecotron staff and in particular Sébastien Devidal for the technical support in running the growth chambers at the Montpellier Ecotron, France. This study benefited from the CNRS human and technical resources allocated to the ECOTRONS Research Infrastructure as well as from the state allocation "Investissement d'Avenir" AnaEE-France ANR-11-INBS-0001. Gas flux measurements were conducted at the platform d'Analyses Chimiques en Ecologie, LabEx Center Méditerranéen de l'Environnement et de la Biodiversité, at CEFE Montpellier, France.

## Author contributions

S.H., N.F., and A.M. had the original idea for the experiment and B.B. provided knowledge and experience for gas flux measurements. L.G. set up the experiment and conducted the laboratory work with the assistance of C.P. L.G. performed the statistical analyses and manuscript redaction with extensive input from S.H., N.F., and A.M.

## Competing interests

The authors declare no competing interests.
