## [Peer Review File · Communications Biology]

Reviewers' comments:

Reviewer #1 (Remarks to the Author):

I reviewed the manuscript number COMMSBIO-20-0044-T on the title: "Higher tree diversity increases soil microbial resistance to drought". The manuscript is well written and it also considers a very important topic, with high relevance and not many studies on it. Although this dynamics have been investigated before, it is the first time that very different ecosystems are compared together, which is the strength of the manuscript. I was very intrigued by the way authors approached the experimental set up and also the statistical analysis. While I think it is positive to try to use such a tailored approach it is also more difficult to set and to justify the specific implementation. Therefore I have many comments on the materials and methods section. For this reason I cannot recommend publication unless my comments are answered appropriately. I am happy to review the manuscript a second time in the case.

More detailed comments:

ABSTRACT

L. 10 Should you just refer to N₂O release? Denitrification refer to the reduction of NO₃, ultimately to N₂, with the release of intermediates. Just a suggestion. Or maybe you could put in parenthesis (N₂O) to make it clear

INTRODUCTION

Literally, one of the best introduction I have seen in many years. Very well done! I am really happy and I have only little suggestion to improve it.

L.36 -37 Here is not clear to which effect you refer to: initially I thought it referred to the resistance and resilience of the microbes, but that would not make sense by the citation. Maybe just change the words "these effects" with something like "plants effect on soil properties...".

L.38 Perhaps change "same" to "specific", I think it would read better

RESULTS

L.114 Is it "Cycle 2" a typo in Table 1? Also the whole interpretation of Table 1 is very hard to read: First at all by using "dredge" you should select the most parsimonious model, which already exclude some factors (indicated by the dash line), so wouldn't make more sense to keep all the factors tested in the first column? For example where did number of trees go?

Second: If time is an individual factor, the interaction of time per DRW treatment should be represented by one value, right?

DISCUSSION

L.212-215 See my comments about how you calculated your cumulative values. With the low resolution during the drought period and especially during the rewetting, this claim is not supported by evidence.

L.218-220 Ok now I see that you try to explain my previous comment. Yet I would really be cautious in this whole paragraph, by trying to extrapolate general conclusion from your experiment. I would strongly suggest to tone down the whole section and to shorten it.

L.264-267 for your discussion see also <https://doi.org/10.1016/j.soilbio.2018.05.027> and <https://doi.org/10.1016/j.soilbio.2018.09.026> and <https://doi.org/10.1016/j.soilbio.2017.04.020>

MATERIALS AND METHODS

I find it a very interesting approach to create a laboratory incubation that is tailored for each specific site. While it is arguable its generalization and approach, I think that authors took very well care in establishing a well thought laboratory incubation (although I have some questions, see below). It would be extremely interesting to compare this to a more standardized laboratory incubation where all sites received the same treatment. What I find somehow striking is the buffer period. While is great that authors had an acclimation period, the strong difference in this buffer period make me sceptical.

As authors state throughout the paper, response of microbial communities to stress are dependent on soil properties, including substrate (or resources) availability. Therefore the replicability between sites might be compromised, given that in absence of growing plants and litter, soil microorganisms might be depleted in primary labile resources, normally available in the field, at different rates between each site. Authors offer as justification: "In order to avoid confounding experimental durations", but the justification doesn't really seem driven by a biological background but rather more a logistical reason. L.458-459 Authors say that *Picea abies* and *Quercus robur* in Białowieża were only replicated once, and that all other mono culture species were replicated twice. However at that site they have 6 monospecific plots and 5 species listed. This would mean that 3 species have two replicates (6plots) and the remaining 2 species (*Picea abies* and *Quercus robur*) would have 2 plots. This gives a total of 8 plots.

Perhaps authors should give a full list of plots, indicating what species was the dominant, number of replicates and also percentages of abundance of each species.

L.472-506 All this paragraph should be moved in the statistic section

L.510-511 Why did you choose 80% WHC? Normally a value of 60% WHC is chosen as this represents (generally) an optimal value for microbial activity

L.529 I think there is a typo here. Should read "been".

L.539-541 Did I understand correctly that the number of days used for drought were calculated from the number of days per year when each specific site reached a certain water level? Then is your experiment testing a double than "normal" number of days if you have two DRW cycles? Or did you split the average number of dry days by the two DRW cycles? Please specify. Also, in your supplementary figure 3 you have 8 different groups, with different duration of drying phase. How did you get to those numbers? And which soils are included in which groups?

L.545-546 How is this possible? It took you about 2 days for the CO₂ and N₂O measurements from what you wrote (L.567-573). Therefore if you start measuring once the microcosm reached the final drying values they would stay about 2 days during the measurements. Maybe you can rephrase this a little to make this point clear.

L.580-581 I do not see how you can generate reliable cumulative values. You explicitly say that you avoid to sample during the so called birch effect (L.563-564). However the birch effect has a highly strong temporal dynamic. If you miss this it is very hard to get proper estimate of cumulative respiration. I would definitely avoid using cumulative respiration when you have only 4 time points measurements over a period of about 130 days. Also by generative cumulative values with such a low number of time points measured you will inflate results from those sites were a higher number of replicates are used (especially because then each site and treatment had 6 technical replicates).

STATISTICS: I find it a bit hard to justify the use in the model of the PC axes. What is the actual rationale behind?

L.622-623 Shouldn't you test normality assumption on the residual and only apply transformation if the assumption are not satisfied? See for example: Kozak and Piepho, 2016 What's normal anyway? Residual plots are more telling than significance tests when checking ANOVA assumptions

L. 626-628 Ok, so you ran an ANCOVA. Now is clearer why you used FDis + CWM + Soil PCA axis in the model. I am not against it, and I think is a valid and nice way to explore data from such different part of the world. However, during an ANCOVA you have extra assumptions (compared to linear model or ANOVA), specifically:

- For each independent variable, the relationship between the dependent variable (y) and the

covariate (x) is linear

- The lines expressing these linear relationships are all parallel (homogeneity of regression slopes)
- The covariate is independent of the treatment effects (i.e. the covariate and independent variables are independent)

From what you described we do not know if these assumptions were met and therefore whether the model is valid. Also, have you tried to use more simple data to run your ancova? For instance just pH or clay content, rather than plot it on a PCA and then use the PCA axis. Somehow using PCA axis (while you generalize and synthesize multiple variables) could hinder the true nature of your variability. Also, how do you justify that for some variables one was significant and therefore used (e.g. root FDis for CO₂ flux resistance) and for other a different one was used (e.g. soil parameter for N₂O flux resistance)?

Reviewer #2 (Remarks to the Author):

The authors tested how tree diversity impacted microbial resistance and recovery under drought conditions, using a soil incubation experiment. This is an well-designed and unique study, that will contribute to our understanding of the connections between plant diversity and microbially driven biogeochemical cycling. One strength of this work is that the authors tracked carbon dioxide flux and nitrous oxide flux across two drying and rewetting cycles, which enables them to track the microbial responses across multiple drying events. Another strength is the use of samples from four sites from a large geographic range and tailoring treatments to the site-specific climate characteristics, likely making the results more ecologically relevant.

While I commend the authors for tailoring the experimental design to climate characteristics of the areas they collected from, I am concerned by the lack of discussion of differences among sites. I want to make sure that the differences in gas responses to drying aren't driven primarily by the site with the longest drought treatment (Colline Metallifere). Ideally, in addition to the analysis in the text, the authors would analyze the data from each site individually and present the tables in the supplementary information and reference how DRW and diversity treatments differed across sites in the text. I think this would greatly strengthen the argument that the relationships between tree diversity and microbial resistance is "robust and consistent."

This paper would benefit from a more thorough introduction and more in-depth explanation of response variables in the results section. More completely introducing the importance of the concepts tested in this experiment in the introduction will smooth the transition from introduction to results without a methods section in between. Similarly, explaining response variables in the results section will help the reader understand the results without referencing the methods first. I elaborate on my recommendations in section and line comments.

Introduction:

I recommend that the introduction be substantially lengthened. Currently, the introduction has information that is too general to fully grasp the importance of the experiment that the authors did. The most important suggestion I have is for the authors to tie the introduction to the variables and treatments they tested. This is important 1. For setting up the results section without a methods section in between, and 2. For making the argument that the experiment they decided to do is

important. While the authors do this for plant diversity (lines 40-47), they do not discuss root traits or soil properties with enough depth. Lastly, the authors should argue for the importance of measuring CO₂ and N₂O.

Line 19: This paper could start stronger by being more specific about where or how much "climate change models predict increase drought frequency and severity". A statement about forests across the world or land in Europe would make this statement stronger.

Line 33-35: The authors should provide more context as to how soil parameters and resource availability are connected to soil microbial resistance and recovery. This is where the authors can argue for the importance of the variables they measured for understanding soil microbial resistance and recovery.

Line 37-40: This sentence on microbial ecology seems out of place given the rest of the paragraph mostly discusses soil and plant diversity effects on soil microbial resistance and recovery. Removing or moving this sentence would make the argument flow better and be more easily understood.

Lines 48-58: Including root traits and soil parameters in this paragraph will prepare the reader for these concepts in the results section.

Line 54: "leaching" is only mentioned one other time in the introduction and not again until the discussion. The authors should introduce the importance of leaching and, either in the introduction or the results, tie the variables used with their connection with leaching.

Results:

Line 69: The transition to discussing root traits is very abrupt. I suggest starting a new paragraph to talk about root traits or add in some guidance for the reader, such as starting the sentence with "In addition to tree species diversity, we also measured root trait effects..."

Lines 71-76: This is a really clear and concise explanation of how root traits impacted CO₂ fluxes.

Table 1: The DOC and TDN can be switched so that the left column in this table is always summarizing Carbon effects and the right column is always summarizing nitrogen effects. If the authors want the table to be more symmetrical, I would recommend moving qCO₂ down to the last row.

line 136-8: This is a clear explanation of this variable.

Line 146: Does 1 indicate 100% recovery in relation to control values or previous sample values? Please clarify in the text.

Line 163: Because this is the first mention of metabolic quotient in the paper, a brief description of metabolic quotient should be added, including how it was calculated.

Line: 163: How much were changes in metabolic quotient driven by changes in microbial biomass or changes in CO₂? To answer this question, I recommend reporting results from tests on microbial biomass. Discussing metabolic quotient in terms of changes of microbial biomass and/or CO₂ would help make the argument in the discussion (lines 241-249) stronger.

Line 168: Nice explanation of soil parameters results.

Line 169: Authors should provide an explanation for acquisitive root traits as they did with conservative root traits in lines 75-76.

Figure 4: The configuration of Figure 4 should match the changes to the bottom half of Table 1.

Discussion:

Line 192-197: "Robust and consistent" is a very strong statement for these results. In addition to my earlier comment, I think there needs to be an acknowledgement of the lack of tree species diversity effect on CO₂ cumulative flux and flux recovery.

Line 207-208: "but subsequence less productivity" doesn't make sense to me. This may be more clear if the authors split this sentence into two sentences.

Line 228: Should "absorptive" root traits be changed to "acquisitional" root traits? Also, I think it would be useful to add a quick description of each here.

Line 243-244: How does metabolic quotient suggest higher microbial stress? I think this is an interesting argument, but needs to be explained more thoroughly.

Line 245-247: The authors measure many of the variables important to this argument, such as microbial biomass and leaching, but don't integrate their results into the discussion of microbial stress response.

Materials and methods

Line 468-469: How close were these 5 soil cores taken from each other? How large was the area sampled? Were the samples taken randomly within an area or spaced out evenly?

Line 474: Authors should explain root trait axes and use the same terminology that is used in the results section (conservative vs acquisitional).

Line 475-486: I am confused about where the data for root traits came from. In line 475, the authors say that root traits were measured from soil cores from each plot, but in line 484, the authors say they use country level species trait means. Please clarify in the text.

Line 498: This sentence confuses me. How was the single FDis value per plot determined? Was FDis averaged across sampling points in the plot?

Line 499: At what spatial scale were soil properties determined? At the plot level or the site level?

Line 528-9: Authors need to include a citation for soil drying. If it's not published a communication reference would work well.

Supplementary Information

This paper would benefit from having a supplemental table with summary data at the plot level. This could include at least mean and standard deviations (when applicable) for root traits (FDis and CWM) and soil properties PCA value.

Reviewer #3 (Remarks to the Author):

The MS entitled "Higher tree diversity increase soil microbial resistance to drought" is a large-scale study about the effect of drought on soil microbial activity in soils of forests collected from different European countries. The effect of drought was investigated by imposing two drying-rewetting cycles. The authors demonstrated that i) microbial aerobic respiration as well as denitrification declined under drought and ii) mixed forest positively impact on microbial community resistance to the stressful conditions

Overall the manuscript is very interesting its novelty is focused on the large-scale experimental approach applied on microcosm from different European sites. The text is clear and well-written also for the not expert scientist.

The results provided support the conclusions

Considering the complex experimental set up used, the statistical analysis is appropriated and well-conducted.

Few points should be better addressed in the manuscript:

1-it is not fully clear why the authors harvested soils at 10 cm deep. Is it a typical procedure?

2- It is clear that for the overall aim of the work, DOC and TOC fractions determination are sufficient to monitor the effect of organic carbon in the different site under the stressful conditions considered. However, it is known the composition of organic compounds (mainly from plants, e.g. root exudates) impact on the soil microbial activity. Did the authors determine the composition of organic C (organic acids, phenols...) in the different soils considered?

3- About the composition of soils harvested, did the authors have information about the content of other elements like heavy metals or other pollutants? Such factors would add further complexity to the experimental system, therefore it would be important to know if all the soils considered are not altered by the presence of such compounds.

Response to referees for the article:

“Higher tree diversity increases soil microbial resistance to drought” (COMMSBIO-20-0044-T)

Dear Reviewers,

We found your comments to be very useful in improving our manuscript, and we are very appreciative of your time spent for the assessment of our work. We have revised the points in question, added more details when needed, and tracked all changes in the manuscript.

Please find our responses to your comments in dark green below

Cordially,
The authors of article COMMSBIO-20-0044-T

Reviewers' comments:

Reviewer #1 (Remarks to the Author):

I reviewed the manuscript number COMMSBIO-20-0044-T on the title: "Higher tree diversity increases soil microbial resistance to drought". The manuscript is well written and it also considers a very important topic, with high relevance and not many studies on it. Although this dynamics have been investigated before, it is the first time that very different ecosystems are compared together, which is the strength of the manuscript. I was very intrigued by the way authors approached the experimental set up and also the statistical analysis. While I think it is positive to try to use such a tailored approach it is also more difficult to set and to justify the specific implementation. Therefore I have many comments on the materials and methods section. For this reason I cannot recommend publication unless my comments are answered appropriately. I am happy to review the manuscript a second time in the case.

More detailed comments:

We appreciate the overall favorable assessment of our work and its relevance by Rev. 1. The implementation of the experimental protocol and the choices to be made were indeed subject of in-depth discussions in our research group. We tried our best to present these choices as clearly as possible, but we understand that some questions remained to which we will answer in detail below.

ABSTRACT

L. 10 Should you just refer to N₂O release? Denitrification refer to the reduction of NO₃, ultimately to N₂, with the release of intermediates. Just a suggestion. Or maybe you could put in parenthesis (N₂O) to make it clear

What we measured was indeed the denitrification potential and not the production of N₂O since we inhibited the transformation of N₂O into N₂. Adding "(N₂O)" would be misleading since the value includes the portion of N that would be transformed into N₂ but is still in the form of N₂O. To avoid confusion, we underlined this in the introduction (line 69).

INTRODUCTION

Literally, one of the best introduction I have seen in many years. Very well done! I am really happy and I have only little suggestion to improve it.

Thank you for this kind and encouraging compliment.

L.36 -37 Here is not clear to which effect you refer to: initially I thought it referred to the resistance and resilience of the microbes, but that would not make sense by the citation. Maybe just change the words "these effects" with something like "plants effect on soil properties...".

Line 46-47 This suggestion has been adopted.

L.38 Perhaps change "same" to "specific", I think it would read better

Line 53 This suggestion has been adopted.

RESULTS

L.114 Is it "Cycle 2" a typo in Table 1? Also the whole interpretation of Table 1 is very hard to read:

Line 783 (Table 1) This was to underline that these results were the change between the first and second cycle, but we agree that it would be clearer as just “Cycle”.

First at all by using “dredge” you should select the most parsimonious model, which already exclude some factors (indicated by the dash line), so wouldn’t make more sense to keep all the factors tested in the first column? For example where did number of trees go?

Line 783 (Table 1) This has been clarified.

Second: If time is an individual factor, the interaction of time per DRW treatment should be represented by one value, right?

Line 783 (Table 1) The time factor was represented by the five ‘stages’ of the experiment (Drought 1, Rewetting 1, Drought 2, and Rewetting 2) and therefore was not a continuous value.

DISCUSSION

L.212-215 See my comments about how you calculated your cumulative values. With the low resolution during the drought period and especially during the rewetting, this claim is not supported by evidence.

We agree with Rev. 1 that we stretched the interpretation a bit too much and that these values need to be discussed more cautiously. We addressed this with the suggestions of the next comment.

L.218-220 Ok now I see that you try to explain my previous comment. Yet I would really be cautious in this whole paragraph, by trying to extrapolate general conclusion from your experiment. I would strongly suggest to tone down the whole section and to shorten it.

Line 207-210, 222-223 We removed extrapolations and better specified the limits of the cumulative values. We also shortened this text passage.

L.264-267 for your discussion see also <https://doi.org/10.1016/j.soilbio.2018.05.027> and <https://doi.org/10.1016/j.soilbio.2018.09.026> and <https://doi.org/10.1016/j.soilbio.2017.04.020>

Thank you for these references, they were very useful and were used in the paper (Lines 230-234, 249-250, and 264).

MATERIALS AND METHODS

I find it a very interesting approach to create a laboratory incubation that is tailored for each specific site. While it is arguable its generalization and approach, I think that authors took very well care in establishing a well thought laboratory incubation (although I have some questions, see below). It would be extremely interesting to compare this to a more standardized laboratory incubation where all sites received the same treatment. What I find somehow striking is the buffer period. While is great that authors had an acclimation period, the strong difference in this buffer period make me sceptical. As authors state throughout the paper, response of microbial communities to stress are dependent on soil properties, including substrate (or resources) availability. Therefore the replicability between sites might be compromised, given that in absence of growing plants and litter, soil microorganisms might be depleted in primary labile resources, normally available in the field, at different rates between each site. Authors offer as justification:” In order to avoid confounding experimental durations”, but the justification doesn’t really seem driven by a biological background but rather more a logistical reason.

There are a multitude of ways to construct this type of experiment and all have their advantages and drawbacks. We ultimately had to make a choice as to what would make more sense biologically considering site variation in drought history and microbial activity duration.

Had we excluded the buffer period it could be criticized that microorganisms active for 14 days were analyzed next to microorganisms active for 130 days. We believe that this would pose even greater problems due to differences in the period of resource exhaustion. We considered it important to keep the microorganisms from all sites active over the same duration irrespective of the exact site-specific duration of drought events to be able to compare microbial responses across sites. Had we split the buffer period between the two cycles, it could be criticized that the DRW cycles were not consecutive (rewetting period length having a significant influence on microbial responses to future DRW; Yu et al 2014). Logistically it would have been much easier to have the end of the second drought of the experimental groups spaced out, but we did not feel that this made sense biologically.

The inclusion of control microcosms helps to reduce potential problems of substrate exhaustion by comparing DRW treatments to parallel controls oppose to microbial activities at the beginning of the experiment (before the drought). By comparing the drought treatments to control microcosms following exactly the same dynamics of potential resource exhaustion, we avoided confounding of drought responses with different levels of resource exhaustion. Actually, the CO₂ flux data of control microcosms presented in Figure 1 do not indicate resource exhaustion with fluxes during the “rewetting 2” period towards the end of the experiment being rather higher than at the beginning of the experiment.

Yu, Z., Wang, G. & Marschner, P. Drying and rewetting - Effect of frequency of cycles and length of moist period on soil respiration and microbial biomass. *Eur. J. Soil Biol.* **62**, 132–137 (2014).

L.458-459 Authors say that *Picea abies* and *Quercus robur* in Białowieża were only replicated once, and that all other mono culture species were replicated twice. However at that site they have 6 monospecific plots and 5 species listed. This would mean that 3 species have two replicates (6plots) and the remaining 2 species (*Picea abies* and *Quercus robur*) would have 2 plots. This gives a total of 8 plots.

Lines 330-331 Thank you for catching this, we forgot to add that *Betula pendula* had no mono-specific plot in Białowieża. This has now been added.

Perhaps authors should give a full list of plots, indicating what species was the dominant, number of replicates and also percentages of abundance of each species.

This table has been added to the supplementary material (Supplementary Table 2).

L.472-506 All this paragraph should be moved in the statistic section

Lines 464-507 It has been moved.

Lines 508-517 We also moved the paragraph on the soil PCA since we found it best fit alongside the paragraph on the CWM PCA.

L.510-511 Why did you choose 80% WHC? Normally a value of 60% WHC is chosen as this represents (generally) an optimal value for microbial activity

Lines 361-363 An explanation was added to the manuscript: We chose 80% to ensure soils were entirely and homogeneously humid; very sandy soils with a low WHC, such as those from the Polish site, were not completely wetted at the typically chosen 60% WHC.

L.529 I think there is a typo here. Should read “been”.

Lines 377-379 This was indeed a typo, thank you for catching this. The sentence was heavily altered, and this typo was removed.

L.539-541 Did I understand correctly that the number of days used for drought were calculated from the number of days per year when each specific site reached a certain water level? Then is your experiment testing a double than “normal” number of days if you have two DRW cycles? Or did you split the average number of dry days by the two DRW cycles? Please specify. Also, in your supplementary figure 3 you have 8 different groups, with different duration of drying phase. How did you get to those numbers? And which soils are included in which groups?

Lines 377-379 A sentence was added to explain how soil drying speed was measured and how this served to define the 8 different groups based on specific drying dynamics. The group number was also added to the new table in the supplementary table to show which soils (i.e. plots) were included in each group (Supplementary Table 2), and a sentence was added to further clarify this (Lines 409-410). Indeed, with the repetitive drought treatment we simulated twice the average “normal” number of days of drought occurrence at each individual site, this has been clarified in the text (Lines 391-391).

L.545-546 How is this possible? It took you about 2 days for the CO₂ and N₂O measurements from what you wrote (L.567-573). Therefore if you start measuring once the microcosm reached the final drying values they would stay about 2 days during the measurements. Maybe you can rephrase this a little to make this point clear.

Lines 496-497 and 399 This is true. This has been rephrased.

L.580-581 I do not see how you can generate reliable cumulative values. You explicitly say that you avoid to sample during the so called birch effect (L.563-564). However the birch effect has a highly strong temporal dynamic. If you miss this it is very hard to get proper estimate of cumulative respiration. I would definitely avoid using cumulative respiration when you have only 4 time points measurements over a period of about 130 days. Also by generative cumulative values with such a low number of time points measured you will inflate results from those sites were a higher number of replicates are used (especially because then each site and treatment had 6 technical replicates).

We agree that the cumulative values must be interpreted cautiously. The caveats associated with this variable have been outlined both in the material and methods (Lines 440-443) as well as in the discussion (Lines 207-210). Since the number of available field plots dictated the number of replicate plots per site, this causes difficulties in the interpretation of site differences if sites were compared explicitly, which we avoided doing.

STATISTICS: I find it a bit hard to justify the use in the model of the PC axes. What is the actual rationale behind?

We understand the reviewer's reservation about our approach because a multivariate dataset like ours can be analysed in several different ways. However, there are multiple reasons for our choice of using the PC axes for the soil descriptors and root traits instead of testing all (or a subset) of these variables. First, we had a large number of collinear soil and root variables. Collinearity leads to high variance inflation, meaning that it is difficult to impossible to assess accurately the contribution of predictors to a model. Therefore, testing all of these variables simultaneously or one by one would have not resolved the problem that they are collinear and even if one or several of them eventually stood out from a significance point of view, it is still difficult to assert with a high degree of confidence that we identified the actual driver (knowing that the other variables are highly correlated). Second, such an approach also increases the type-II errors (false positives) due to testing a much higher number of predictors. Therefore, in order to deal with these issues, we used a well-established method (see e.g. Doemann et al. 2013) of dealing with collinearity in our dataset.

Dormann, C. F., Elith, J., Bacher, S., Buchmann, C., Carl, G., Carré, G., ... Lautenbach, S. (2013). Collinearity: A review of methods to deal with it and a simulation study evaluating their performance. *Ecography*, 36(1), 27–46. <https://doi.org/10.1111/j.1600-0587.2012.07348.x>

L.622-623 Shouldn't you test normality assumption on the residual and only apply transformation if the assumption are not satisfied? See for example: Kozak and Piepho, 2016 What's normal anyway? Residual plots are more telling than significance tests when checking ANOVA assumptions

Lines 538-541 We have elaborated on this. We did test normality assumptions on the residuals and applied transformations when assumptions were not filled (normality distribution, homogeneity, and probability Q-Q plot).

L. 626-628 Ok, so you ran an ANCOVA. Now is clearer why you used FDis + CWM + Soil PCA axis in the model. I am not against it, and I think is a valid and nice way to explore data from such different part of the world. However, during an ANCOVA you have extra assumptions (compared to linear model or ANOVA), specifically:

- For each independent variable, the relationship between the dependent variable (y) and the covariate (x) is linear
- The lines expressing these linear relationships are all parallel (homogeneity of regression slopes)
- The covariate is independent of the treatment effects (i.e. the covariant and independent variables are independent)

From what you described we do not know if these assumptions were met and therefore whether the model is valid.

Lines 538-541 We did verify that these assumptions were met but did not mention this in the materials and methods. This had been added now. However, we would like to emphasize that our statistical models are not typical ANCOVAs because this implies a combination of factorial and

continuous predictors and no random factors. Instead, we used a generalized mixed-effects approach with random intercepts for each site, but with the assumption of homogeneity of slopes at the site level.

Also, have you tried to use more simple data to run your ancova? For instance just pH or clay content, rather than plot it on a PCA and then use the PCA axis. Somehow using PCA axis (while you generalize and synthetise multiple variables) could hinder the true nature of your variability.

We tried this and whether we included soil pH or clay for the most part didn't change the results in comparison with the Soil PC1 axis. On occasion though, the soil variable included had little effect on the flux and instead the tree diversity variable became significant. Whether we chose to include soil pH or clay (or another soil variable) would then determine whether or not the tree diversity was significant or not (see table below highlighted in yellow). We felt that by using the Soil PC1 better allowed us to impartially include soil variables.

	CO ₂ flux						N ₂ O flux					
	R ² _m = 0.329		R ² _c = 0.467		AIC= 5082.1		R ² _m = 0.147		R ² _c = 0.711		AIC= 385.1	
	Est.	SE	df	t-value	p-value		Est.	SE	df	t-value	p-value	
Tree sp. no.	-	-	-	-	-		-	-	-	-	-	
Tree sp. no.:DRW	-	-	-	-	-		-	-	-	-	-	
Drought 1:DRW	-1.18	0.13	1809.0	-9.06	< 2e-16	***	-0.33	0.04	1787.6	-8.86	< 2e-16	***
Rewetting 1:DRW	0.35	0.13	1809.0	2.67	0.01	**	0.06	0.04	1787.6	1.71	0.09	.
Drought 2:DRW	-1.55	0.13	1809.0	-11.94	< 2e-16	***	-0.36	0.04	1787.6	-9.57	< 2e-16	***
Rewetting 2:DRW	0.39	0.13	1809.0	3.04	2.4E-03	**	0.02	0.04	1787.6	0.42	0.67	
Soil pH	-	-	-	-	-		0.15	0.04	61.6	3.95	2.0E-04	***
Root FDis	-	-	-	-	-		-	-	-	-	-	
Root CWM	0.15	0.04	6.2	3.47	0.01	*	-	-	-	-	-	
	CO ₂ flux						N ₂ O flux					
	R ² _m = 0.335		R ² _c = 0.467		AIC= 5081.9		R ² _m = 0.252		R ² _c = 0.73		AIC= 377.6	
	Est.	SE	df	t-value	p-value		Est.	SE	df	t-value	p-value	
Tree sp. no.	-	-	-	-	-		0.01	0.06	64.8	0.15	0.88	
Tree sp. no.:DRW	-	-	-	-	-		0.06	0.02	1787.0	2.60	0.01	**
Drought 1:DRW	-1.18	0.13	1809.0	-9.06	< 2e-16	***	-0.33	0.04	1787.0	-8.89	< 2e-16	***
Rewetting 1:DRW	0.35	0.13	1809.0	2.67	0.01	**	0.06	0.04	1787.0	1.70	0.09	.
Drought 2:DRW	-1.55	0.13	1809.0	-11.94	< 2e-16	***	-0.36	0.04	1787.0	-9.61	< 2e-16	***
Rewetting 2:DRW	0.39	0.13	1809.0	3.04	2.4E-03	**	0.02	0.04	1787.0	0.41	0.68	
Soil clay	0.01	0.01	10.1	1.73	0.11		-	-	-	-	-	
Root FDis	-	-	-	-	-		-	-	-	-	-	
Root CWM	0.11	0.04	5.8	2.79	0.03	*	-	-	-	-	-	

Also, how do you justify that for some variables one was significant and therefore used (e.g. root FDis for CO₂ flux resistance) and for other a different one was used (e.g. soil parameter for N₂O flux resistance)?

Lines 796-797 (Table 1) All variables listed were used, but not all were retained in the minimal adequate model after running the dredge function. Non-retained variables are indicated by a dash “-” while some variables were retained but not significant. This is perhaps the source of the confusion. We added to the table description to address this.

Reviewer #2 (Remarks to the Author):

The authors tested how tree diversity impacted microbial resistance and recovery under drought conditions, using a soil incubation experiment. This is an well-designed and unique study, that will contribute to our understanding of the connections between plant diversity and microbially driven biogeochemical cycling. One strength of this work is that the authors tracked carbon dioxide flux and nitrous oxide flux across two drying and rewetting cycles, which enables them to track the microbial responses across multiple drying events. Another strength is the use of samples from four sites from a large geographic range and tailoring treatments to the site-specific climate characteristics, likely making the results more ecologically relevant.

While I commend the authors for tailoring the experimental design to climate characteristics of the areas they collected from, I am concerned by the lack of discussion of differences among sites. I want to make sure that the differences in gas responses to drying aren't driven primarily by the site with the longest drought treatment (Colline Metallifere). Ideally, in addition to the analysis in the text, the authors would analyze the data from each site individually and present the tables in the supplementary information and reference how DRW and diversity treatments differed across sites in the text. I think this would greatly strengthen the argument that the relationships between tree diversity and microbial resistance is “robust and consistent.”

Lines 334-338 An explanation as to why we did not explore results at the site-level has been added: The sampling design with the total plot number and their distribution over four forest ecosystems and including a wide range of tree species is strong to address the generality of our hypothesis that microbial responses to DRW cycles are modified by tree species mixing but weak to identify site-specific patterns with plot numbers too limiting within specific sites for robust testing. The statistical power is not sufficient to run the relatively complex models at individual site level. We acknowledge that the wording in the Abstract and in the Discussion was perhaps suggesting that all sites showed the same results when tested individually. We rewrote this differently to insist on the fact that comparing mono-specific forest plots with mixed species forest plots across different ecosystems and including different tree species provided results that are robust to interpret, because of the different sites and conditions embraced in our test.

This paper would benefit from a more thorough introduction and more in-depth explanation of response variables in the results section. More completely introducing the importance of the concepts tested in this experiment in the introduction will smooth the transition from introduction to results without a methods section in between. Similarly, explaining response

variables in the results section will help the reader understand the results without referencing the methods first. I elaborate on my recommendations in section and line comments.

We understand that without reading the details in the M&M section, the transition from the Introduction to the Results was a bit abrupt. However, since the Introduction was particularly appreciated by Rev. 1, we didn't want to modify it too heavily and were seeking a compromise between the different perceptions by the two reviewers. The introduction has been lengthened somewhat to include more explanation of why the chosen variables were included and why we considered them important. We also added short descriptions of the variables in the results.

Introduction:

I recommend that the introduction be substantially lengthened. Currently, the introduction has information that is too general to fully grasp the importance of the experiment that the authors did. The most important suggestion I have is for the authors to tie the introduction to the variables and treatments they tested. This is important 1. For setting up the results section without a methods section in between, and 2. For making the argument that the experiment they decided to do is important. While the authors do this for plant diversity (lines 40-47), they do not discuss root traits or soil properties with enough depth. Lastly, the authors should argue for the importance of measuring CO₂ and N₂O.

We have further developed the importance of soil properties and root traits as well as why we measured CO₂ and N₂O. On the other hand, as explained in the response to the former comment by Rev. 2 we refrained from a "substantial" lengthening to acknowledge Rev. 1's comments. We hope the proposed compromise is satisfying for both reviewers.

Line 19: This paper could start stronger by being more specific about where or how much "climate change models predict increase drought frequency and severity". A statement about forests across the world or land in Europe would make this statement stronger.

Lines 18-20 This sentence has been embellished.

Line 33-35: The authors should provide more context as to how soil parameters and resource availability are connected to soil microbial resistance and recovery. This is where the authors can argue for the importance of the variables they measured for understanding soil microbial resistance and recovery.

Lines 36-45 More information has been added.

Line 37-40: This sentence on microbial ecology seems out of place give the rest of the paragraph mostly discusses soil and plant diversity effects on soil microbial resistance and recovery. Removing or moving this sentence would make the argument flow better and be more easily understood.

Lines 48-52 We added more information about the link between soil, plant diversity, and soil microorganisms to help argument flow.

Lines 52-55 We prefer to keep this sentence (the sentence that you referred to) here since it further develops how the plant community could alter the microbial resistance and/or recovery.

Lines 48-58: Including root traits and soil parameters in this paragraph will prepare the reader for these concepts in the results section.

We chose to add more information about this in the paragraph about soil (Lines 36-45) and then further underlined that these were factors included in our study in the last paragraph of the introduction (Lines 72-74).

Line 54: “leaching” is only mentioned one other time in the introduction and not again until the discussion. The authors should introduce the importance of leaching and, either in the introduction or the results, tie the variables used with their connection with leaching.

We now better touched on the importance of leaching in the introduction (Lines 42-45) and then further developed it in the discussion (Lines 253-255). What we measured was extractable DOC and TDN, which can be cautiously interpreted as an indicator of potential leaching losses but should not be confounded with true leaching.

Results:

Line 69: The transition to discussing root traits is very abrupt. I suggest starting a new paragraph to talk about root traits or add in some guidance for the reader, such as starting the sentence with “In addition to tree species diversity, we also measured root trait effects...”

Line 94 We added the suggested transition.

Lines 71-76: This is a really clear and concise explanation of how root traits impacted CO₂ fluxes.

Thank you

Table 1: The DOC and TDN can be switched so that the left column in this table is always summarizing Carbon effects and the right column is always summarizing nitrogen effects. If the authors want the table to be more symmetrical, I would recommend moving qCO₂ down to the last row.

Line 783 (Table 1) This has been done.

line 136-8: This is a clear explanation of this variable.

Line 146: Does 1 indicates 100% recovery in relation to control values or previous sample values? Please clarify in the text.

Line 139 This has been clarified.

Line 163: Because this is the first mention of metabolic quotient in the paper, a brief description of metabolic quotient should be added, including how it was calculated.

Lines 157-158 This has been added.

Line: 163: How much were changes in metabolic quotient driven by changes in microbial biomass or changes in CO₂? To answer this question, I recommend reporting results from tests on microbial biomass. Discussing metabolic quotient in terms of changes of microbial biomass and/or CO₂ would help make the argument in the discussion (lines 241-249) stronger.

Figures and GLMM results for the basal respiration and microbial biomass (the variables used to calculate qCO₂) were added to the supplementary materials (Supplementary Table 5, Supplementary Fig. 4) and analyzed in the text (Lines 158-163, 242-245).

Line 168: Nice explanation of soil parameters results.

Thank you

Line 169: Authors should provide an explanation for acquisitive root traits as they did with conservative root traits in lines 75-76.

Line 169 This has been done.

Figure 4: The configuration of Figure 4 should match the changes to the bottom half of Table 1.

Lines 777-783 (Figure 4) Good remark! This has been done: the variables DOC is now in the position 'a', TDN is in position 'b', and qCO_2 is in position 'c'.

Discussion:

Line 192-197: “Robust and consistent” is a very strong statement for these results. In addition to my earlier comment, I think there needs to be an acknowledgement of the lack of tree species diversity effect on CO_2 cumulative flux and flux recovery.

Line 180-182 We have elaborated on the lack of tree species diversity effect on cumulative flux and flux recovery. We also removed “consistent” (Lines 176-177, 309) (as well as in the

abstract), because we understand that it could be read as if sites were compared explicitly. As we explained above in our answer to one of the general comments, we lack statistical power to reasonably run these site-level analyses.

Line 207-208: “but subsequence less productivity” doesn’t make sense to me. This may be more clear if the authors split this sentence into two sentences.

Line 190-193 We have shortened the sentence and removed “but subsequent less productivity” to improve clarity.

Line 228: Should “absorptive” root traits be changed to “acquisitional” root traits? Also, I think it would be useful to add a quick description of each here.

Line 212-216: There was probably a misunderstanding. “Absorptive” refers to the type of fine roots studied (i.e. the first three orders of fine roots that are often referred to as “absorptive roots”), while “acquisitional” refers to a strategy of resource acquisition. A description has been added to avoid confusion.

Line 243-244: How does metabolic quotient suggest higher microbial stress? I think this is an interesting argument, but needs to be explained more thoroughly.

Line 235-238 The link between metabolic quotient and microbial stress has been elaborated.

Line 245-247: The authors measure many of the variables important to this argument, such as microbial biomass and leaching, but don’t integrate their results into the discussion of microbial stress response.

Line 238-243, 257-258 More has been added to the text that should now better integrate these different variables.

Materials and methods

Line 468-469: How close were these 5 soil cores taken from each other? How large was the area sampled? Were the samples taken randomly within an area or spaced out evenly?

Line 343-345 This information has been added.

Line 474: Authors should explain root trait axes and use the same terminology that is used in the results section (conservative vs acquisitional).

Line 495-501 This is already included at the end of the paragraph. We are unsure whether Reviewer 2 is referring to something else.

Line 475-486: I am confused about where the data for root traits came from. In line 475, the authors say that root traits were measured from soil cores from each plot, but in line 484, the authors say they use country level species trait means. Please clarify in the text.

Line 472-477, 484-495 We acknowledge that this was not very clear in the previous version of the manuscript. The root traits were measured on the fine roots that came from the root cores we collected in the field, but since we encountered difficulties differentiating between *Quercus* species in certain mixed plots in Italy we were unable to determine these traits at the plot level. We therefore used root trait values at the site level (using the mono-specific plot values), but still reported the site-level root trait values to the plot-level tree species basal areas which allowed us

to still have CWM and FDis values at the plot-level. The explanation has been reworked to be clearer.

Line 498: This sentence confuses me. How was the single FDis value per plot determined? Was FDis averaged across sampling points in the plot?

Line 501-502 The FDis was calculated at the plot level since the five subplot soil samples had been pooled for the experiment. This has been clarified in the text.

Line 499: At what spatial scale were soil properties determined? At the plot level or the site level?

Line 509 At the plot level. This information has been added to the text.

Line 528-9: Authors need to include a citation for soil drying. If it's not published a communication reference would work well.

Line 374-377 We measured this right before the start of the experiment using subsamples of soil which were subsequently excluded from the experiment. This is now better explained in the text.

Supplementary Information

This paper would benefit from having a supplemental table with summary data at the plot level. This could include at least mean and standard deviations (when applicable) for root traits (FDis and CWM) and soil properties PCA value.

The root trait FDis values and soil and root CWM PC1 scores were added to the Supplementary Table 1, as well as the site-level values for the variables used to calculate these indices.

Reviewer #3 (Remarks to the Author):

The MS entitled "Higher tree diversity increase soil microbial resistance to drought" is a large-scale study about the effect of drought on soil microbial activity in soils of forests collected from different European countries. The effect of drought was investigated by imposing two drying-rewetting cycles. The authors demonstrated that i) microbial aerobic respiration as well as denitrification declined under drought and ii) mixed forest positively impact on microbial community resistance to the stressful conditions

Overall the manuscript is very interesting its novelty is focused on the large-scale experimental approach applied on microcosm from different European sites. The text is clear and well-written also for the not expert scientist.

The results provided support the conclusions

Considering the complex experimental set up used, the statistical analysis is appropriated and well-conducted.

We very much appreciate this positive general feedback on our manuscript.

Few points should be better addressed in the manuscript:

1-it is not fully clear why the authors harvested soils at 10 cm deep. Is it a typical procedure?

Line 345-347 A sentence has been added to explain this. This is indeed the typically topsoil sampling depth in soil ecology.

2- It is clear that for the overall aim of the work, DOC and TOC fractions determination are sufficient to monitor the effect of organic carbon in the different site under the stressful conditions considered. However, it is known the composition of organic compounds (mainly from plants, e.g. root exudates) impact on the soil microbial activity. Did the authors determine the composition of organic C (organic acids, phenols...) in the different soils considered?
No, we did not measure this, although this would be indeed an interesting factor to take into consideration.

3- About the composition of soils harvested, did the authors have information about the content of other elements like heavy metals or other pollutants? Such factors would add further complexity to the experimental system, therefore it would be important to know if all the soils considered are not altered by the presence of such compounds.
We do not have information of this type from our study sites, but these are mature, natural forests and it is unlikely that heavy metals or other pollutants are prominent factors influencing these systems.

REVIEWERS' COMMENTS:

Reviewer #1 (Remarks to the Author):

Reviewers' comments:

I reviewed the answers to my previous comments on the manuscript number COMMSBIO-20-0044-T on the title: "Higher tree diversity increases soil microbial resistance to drought".

I am generally very pleased to the extent my questions were taken care of. I have however still some questions, before I can fully recommend submission. I will report some of my previous comments and the author answer, followed by my new comment:

1)

QUESTION: Second: If time is an individual factor, the interaction of time per DRW treatment should be represented by one value, right?

ANSWER: Line 783 (Table 1) The time factor was represented by the five 'stages' of the experiment (Drought 1, Rewetting 1, Drought 2, and Rewetting 2) and therefore was not a continuous value.

NEW QUESTION

Ok now I am quite confused. If you introduce a factor in a model, this normally includes multiple levels (let's say in this case the factor is stage or time, and the levels are the different levels are D1, R1, D2, R2) but should have one p-value. Within this variable you can then test which stage was significant or not with some post-hoc test. Can you please clarify this point?

2)

QUESTION I find it a very interesting approach to create a laboratory incubation that is tailored for each specific site. While it is arguable its generalization and approach, I think that authors took very well care in establishing a well thought laboratory incubation (although I have some questions, see below). It would be extremely interesting to compare this to a more standardized laboratory incubation where all sites received the same treatment. What I find somehow striking is the buffer period. While is great that authors had an acclimation period, the strong difference in this buffer period make me sceptical. As authors state throughout the paper, response of microbial communities to stress are dependent on soil properties, including substrate (or resources) availability. Therefore the replicability between sites might be compromised, given that in absence of growing plants and litter, soil microorganisms might be depleted in primary labile resources, normally available in the field, at different rates between each site. Authors offer as justification: "In order to avoid confounding experimental durations", but the justification doesn't really seem driven by a biological background but rather more a logistical reason.

ANSWER There are a multitude of ways to construct this type of experiment and all have their advantages and drawbacks. We ultimately had to make a choice as to what would make more sense biologically considering site variation in drought history and microbial activity duration.

Had we excluded the buffer period it could be criticized that microorganisms active for 14 days were analyzed next to microorganisms active for 130 days. We believe that this would pose even greater problems due to differences in the period of resource exhaustion. We considered it important to keep the microorganisms from all sites active over the same duration irrespective of the exact site-specific duration of drought events to be able to compare microbial responses across sites. Had we split the buffer period between the two cycles, it could be criticized that the DRW cycles were not consecutive (rewetting period length having a significant influence on microbial responses to future DRW; Yu et al 2014). Logistically it would have been much easier to have the end of the second drought of the experimental groups spaced out, but we did not feel that this made sense biologically.

The inclusion of control microcosms helps to reduce potential problems of substrate exhaustion by comparing DRW treatments to parallel controls oppose to microbial activities at the beginning of the

experiment (before the drought). By comparing the drought treatments to control microcosms following exactly the same dynamics of potential resource exhaustion, we avoided confounding of drought responses with different levels of resource exhaustion. Actually, the CO₂ flux data of control microcosms presented in Figure 1 do not indicate resource exhaustion with fluxes during the “rewetting 2” period towards the end of the experiment being rather higher than at the beginning of the experiment.

Yu, Z., Wang, G. & Marschner, P. Drying and rewetting - Effect of frequency of cycles and length of moist period on soil respiration and microbial biomass. *Eur. J. Soil Biol.* 62, 132–137 (2014).

NEW QUESTION I agree with the answer of the authors that many aspects can be criticised, and there is not a perfect scenario. However, you could include a small discussion about this. What I am more sceptical about is that the different buffer periods (which inevitably exhaust some of the readily available resources) may somehow affect micro-organism response to drying and rewetting (at least the first cycle). In other words, your sentence “In order to avoid confounding experimental durations”, is not enough for a reader to fully understand your choice. I would recommend to put a small discussion about it.

3)

QUESTION L.510-511 Why did you choose 80% WHC? Normally a value of 60% WHC is chosen as this represents (generally) an optimal value for microbial activity

ANSWER Lines 361-363 An explanation was added to the manuscript: We chose 80% to ensure soils were entirely and homogeneously humid; very sandy soils with a low WHC, such as those from the Polish site, were not completely wetted at the typically chosen 60% WHC.

NEW QUESTION: I have some how hard time believing this. Also the explanation seems quite subjective. How did you assess that the soil were not completely wetted? I am stressing this point as I consider it important. However I have found this paper: “Effect of temperature and moisture on rates of carbon mineralization in a Mediterranean oak forest soil under controlled and field conditions”, which actually show that maximum activity was reached at 60% or 80% WHC. This could help supporting your choice. Perhaps you could include this paper to justify your choice.

4)

QUESTION L.580-581 I do not see how you can generate reliable cumulative values. You explicitly say that you avoid to sample during the so called birch effect (L.563-564). However the birch effect has a highly strong temporal dynamic. If you miss this it is very hard to get proper estimate of cumulative respiration. I would definitely avoid using cumulative respiration when you have only 4 time points measurements over a period of about 130 days. Also by generative cumulative values with such a low number of time points measured you will inflate results from those sites were a higher number of replicates are used (especially because then each site and treatment had 6 technical replicates).

ANSWER We agree that the cumulative values must be interpreted cautiously. The caveats associated with this variable have been outlined both in the material and methods (Lines 440-443) as well as in the discussion (Lines 207-210). Since the number of available field plots dictated the number of replicate plots per site, this causes difficulties in the interpretation of site differences if sites were compared explicitly, which we avoided doing.

NEW QUESTION: I am still not sure that it is enough the steps you took to be cautious in considering the cumulative data results. You know that the calculation are not correct without a temporal resolution of the birch effect. If you want to keep the cumulative data in your analysis you should state clearly in your figure that the cumulative values do not represent appropriately the initial rewetting dynamic.

5)

COMPLETELY NEW QUESTION

In line 521 (version with the corrections) you state: "extreme values were removed (± 3 times the IQR)." Did you consider these as outliers? Which IQR did you consider? For each site, DRW cycle.....? Would have the inclusion of these values changed your results some how? How many values were excluded? I know that outlier detection can be critical to many aspect, including the results we obtain. I know that there is no universal test for outliers. What I normally do is to try testing with and without, to really know whether they make a significant difference or not. In other words, I am asking to explain this better as it can appear vague as it is at the moment.

Reviewer #2 (Remarks to the Author):

The authors were very thorough in their responses to my suggestions. I think the detail added in the introduction is interesting and sufficient to introduce the study. The methods section is much clearer now. This is a unique and exciting study that is an important contribution to understanding the importance of plant species diversity in soil microbial resistance to drought. I only have a few remaining comments.

I thank the authors for their response to my comment to include site-specific analysis. I think their explanation and the changes to the text clarify the strength and purpose of the analysis and the conclusion of the study. I do maintain that there needs to be more site-specific information provided in the supplementary materials. I recommend the authors add mean and standard deviation information for CO₂ and N₂O fluxes, DOC, TDN, and microbial biomass summarized at the level of site and species number to Supplementary Table 2.

The authors have thoroughly explained where samples used for the root data came from. However, I'm still not clear what methods they used to measure root traits. Are the methods from Wambsganss et al? Right now for me line 475 reads as though the soil samples were from that study, but not that the root trait measurements are from that study. Please clarify or provide information regarding methods for root trait measurements.

Line 764-765 (Figure 2). There are two mentions of CO₂ fluxes that should be changed to N₂O fluxes.

Response to referees' comments and questions

REVIEWERS' COMMENTS:

Reviewer #1 (Remarks to the Author):

Reviewers' comments:

I reviewed the answers to my previous comments on the manuscript number COMMSBIO-20-0044-T on the title: "Higher tree diversity increases soil microbial resistance to drought". I am generally very pleased to the extent my questions were taken care of. I have however still some questions, before I can fully recommend submission. I will report some of my previous comments and the author answer, followed by my new comment:

1)

QUESTION: Second: If time is an individual factor, the interaction of time per DRW treatment should be represented by one value, right?

ANSWER: Line 783 (Table 1) The time factor was represented by the five 'stages' of the experiment (Drought 1, Rewetting 1, Drought 2, and Rewetting 2) and therefore was not a continuous value.

NEW QUESTION

Ok now I am quite confused. If you introduce a factor in a model, this normally includes multiple levels (let's say in this case the factor is stage or time, and the levels are the different levels are D1, R1, D2, R2) but should have one p-value. Within this variable you can then test which stage was significant or not with some post-hoc test. Can you please clarify this point?

ANSWER: Yes, you are correct, the variable stage with four levels was included in the model as well as the interaction with the drought treatment. However, we opted for a regression type model output ("summary" syntax in R) instead of a anova-type output ("anova" syntax), because it allows to assess the fitted parameter estimates (slopes) for the main effects as well as the interactions (DR1:DRW to R2:DRW). This is arguably a better approach than a posthoc-test as typical posthoc tests do not take into account random effects. We opted to not add the output lines for the stage main effect to an already relatively full table 1, because this information contributes little for further understanding given that when higher level interactions are significant, the main effect outputs of the variables retained in interactions are not interpretable. This is now mentioned in the table legend (lines 827-830).

2)

QUESTION I find it a very interesting approach to create a laboratory incubation that is tailored for each specific site. While it is arguable its generalization and approach, I think that authors took very well care in establishing a well thought laboratory incubation (although I have some questions, see below). It would be extremely interesting to compare this to a more standardized laboratory incubation where all sites received the same treatment. What I find somehow striking is the buffer period. While is great that authors had an acclimation period,

the strong difference in this buffer period make me sceptical. As authors state throughout the paper, response of microbial communities to stress are dependent on soil properties, including substrate (or resources) availability. Therefore the replicability between sites might be compromised, given that in absence of growing plants and litter, soil microorganisms might be depleted in primary labile resources, normally available in the field, at different rates between each site. Authors offer as justification:” In order to avoid confounding experimental durations”, but the justification doesn’t really seem driven by a biological background but rather more a logistical reason.

ANSWER There are a multitude of ways to construct this type of experiment and all have their advantages and drawbacks. We ultimately had to make a choice as to what would make more sense biologically considering site variation in drought history and microbial activity duration.

Had we excluded the buffer period it could be criticized that microorganisms active for 14 days were analyzed next to microorganisms active for 130 days. We believe that this would pose even greater problems due to differences in the period of resource exhaustion. We considered it important to keep the microorganisms from all sites active over the same duration irrespective of the exact site-specific duration of drought events to be able to compare microbial responses across sites. Had we split the buffer period between the two cycles, it could be criticized that the DRW cycles were not consecutive (rewetting period length having a significant influence on microbial responses to future DRW; Yu et al 2014). Logistically it would have been much easier to have the end of the second drought of the experimental groups spaced out, but we did not feel that this made sense biologically.

The inclusion of control microcosms helps to reduce potential problems of substrate exhaustion by comparing DRW treatments to parallel controls oppose to microbial activities at the beginning of the experiment (before the drought). By comparing the drought treatments to control microcosms following exactly the same dynamics of potential resource exhaustion, we avoided confounding of drought responses with different levels of resource exhaustion. Actually, the CO₂ flux data of control microcosms presented in Figure 1 do not indicate resource exhaustion with fluxes during the “rewetting 2” period towards the end of the experiment being rather higher than at the beginning of the experiment.

Yu, Z., Wang, G. & Marschner, P. Drying and rewetting - Effect of frequency of cycles and length of moist period on soil respiration and microbial biomass. *Eur. J. Soil Biol.* 62, 132–137 (2014).

NEW QUESTION I agree with the answer of the authors that many aspects can be criticised, and there is not a perfect scenario. However, you could include a small discussion about this. What I am more sceptical about is that the different buffer periods (which inevitably exhaust some of the readily available resources) may somehow affect micro-organism response to drying and rewetting (at least the first cycle). In other words, your sentence “In order to avoid confounding experimental durations”, is not enough for a reader to fully understand your choice. I would recommend to put a small discussion about it.

We agree that it is good idea to further elaborate this for the reader. More has been added to the text (lines 406-416):

In order to ensure the microbial communities were active for the same duration, which better allows analysis of microbial activity from different sites, we staggered the beginning of the first drying-rewetting cycle so that all microcosms finished the two DRW cycle treatment at

approximately the same time. We refer to the period between the first gas measurement and the beginning of the first drying period as the buffer period (Supplementary Fig. 3). Although the buffer period could have led to an exhaustion of some of the readily available resources for microorganisms, comparing the activity of the DRW treatment microcosms to parallel control microcosms following exactly the same dynamics of potential resource exhaustion should have avoided confounding of drought responses with different levels of resource exhaustion.

3)

QUESTION L.510-511 Why did you choose 80% WHC? Normally a value of 60% WHC is chosen as this represents (generally) an optimal value for microbial activity

ANSWER Lines 361-363 An explanation was added to the manuscript: We chose 80% to ensure soils were entirely and homogeneously humid; very sandy soils with a low WHC, such as those from the Polish site, were not completely wetted at the typically chosen 60% WHC.

NEW QUESTION: I have some how hard time believing this. Also the explanation seems quite subjective. How did you assess that the soil were not completely wetted? I am stressing this point as I consider it important. However I have found this paper: “Effect of temperature and moisture on rates of carbon mineralization in a Mediterranean oak forest soil under controlled and field conditions”, which actually show that maximum activity was reached at 60% or 80% WHC. This could help supporting your choice. Perhaps you could include this paper to justify your choice.

Since the microcosms were clear vials, we were able to see by the color change of wetted soil that when 60% WHC of water had been added, the soil at the bottom of the vial was still dry after 24 hours (this was noticed during pre-experiment testing).

Thank you for the suggested paper, it has been added to the text to help justify our choice (lines 363-364):

Maximum microbial mineralization activity appears to be reached between 60% and 80% WHC⁵⁵. We chose 80% to ensure soils were entirely and homogeneously humid; very sandy soils with a low WHC, such as those from the Polish site, were not completely wetted at the typically chosen 60% WHC.

⁵⁵ Rey, A., Petsikos, C., Jarvis, P. G. & Grace, J. Effect of temperature and moisture on rates of carbon mineralization in a Mediterranean oak forest soil under controlled and field conditions. *Eur. J. Soil Sci.* 56, 589–599 (2005).

4)

QUESTION L.580-581 I do not see how you can generate reliable cumulative values. You explicitly say that you avoid to sample during the so called birch effect (L.563-564). However the birch effect has a highly strong temporal dynamic. If you miss this it is very hard to get proper estimate of cumulative respiration. I would definitely avoid using cumulative respiration when you have only 4 time points measurements over a period of about 130 days. Also by generative cumulative values with such a low number of time points measured you will inflate results from those sites were a higher number of replicates are used (especially because then each site and treatment had 6 technical replicates).

ANSWER We agree that the cumulative values must be interpreted cautiously. The caveats associated with this variable have been outlined both in the material and methods (Lines 440-443) as well as in the discussion (Lines 207-210). Since the number of available field plots dictated the number of replicate plots per site, this causes difficulties in the interpretation of site differences if sites were compared explicitly, which we avoided doing.

NEW QUESTION: I am still not sure that it is enough the steps you took to be cautious in considering the cumulative data results. You know that the calculation are not correct without a temporal resolution of the birch effect. If you want to keep the cumulative data in your analysis you should state clearly in your figure that the cumulative values do not represent appropriately the initial rewetting dynamic.

We have added more to the figure legend to further caution the reader on the calculation and interpretation of the cumulative values (lines 778-779 and 791-792):
...calculated cumulative values are only rough estimates because our measurements did not cover the initial rewetting dynamics.

5)

COMPLETELY NEW QUESTION

In line 521 (version with the corrections) you state: “extreme values were removed (± 3 times the IQR).” Did you consider these as outliers? Which IQR did you consider? For each site, DRW cycle.....? Would have the inclusion of these values changed your results some how? How many values were excluded? I know that outlier detection can be critical to many aspect, including the results we obtain. I know that there is no universal test for outliers. What I normally do is to try testing with and without, to really know whether they make a significant difference or not. In other words, I am asking to explain this better as it can appear vague as it is at the moment.

We considered the IQR across all treatments and sites for the entire data set as, according to common protocols of outlier identification We added “ ± 3 times the IQR of all values for each variable” (lines 538-539) to clarify this.

Yes, we considered them outliers and the number of points removed never exceeded 11% of the total number of points per group. A group was defined according to the variable: individual CO₂ and N₂O flux variables were grouped by tree species number, treatment, and experimental stage; cumulative CO₂ and N₂O fluxes, DOC, TDN, and qCO₂ variables were grouped by tree species number and treatment; and the CO₂ and N₂O resistance and recovery indices were grouped by tree species number. Ungrouped, the number of removed values never exceeded 8% of the total values.

We performed our analyses with and without outlying variables, and the inclusion of these values pulling variable tendencies in a direction that was not representative of non-outlying values. After considerable testing and discussion, we concluded that it was best to remove these values and chose ± 3 times the IQR as the most unbiased method of determining a threshold.

Reviewer #2 (Remarks to the Author):

The authors were very thorough in their responses to my suggestions. I think the detail added in the introduction is interesting and sufficient to introduce the study. The methods section is much clearer now. This is a unique and exciting study that is an important contribution to understanding the importance of plant species diversity in soil microbial resistance to drought. I only have a few remaining comments.

I thank the authors for their response to my comment to include site-specific analysis. I think their explanation and the changes to the text clarify the strength and purpose of the analysis and the conclusion of the study. I do maintain that there needs to be more site-specific information provided in the supplementary materials. I recommend the authors add mean and standard deviation information for CO₂ and N₂O fluxes, DOC, TDN, and microbial biomass summarized at the level of site and species number to Supplementary Table 2.

The CO₂ and N₂O fluxes, DOC, TDN, and microbial biomass data would not be interpretable at just the site and tree species number level, the data would also need to be differentiated by treatment and, depending on the variable, experimental stage. This would be too much information to include in the supplementary materials document. We therefore included it in the supplementary material excel document and cited it within the main text (lines 480-482).

The authors have thoroughly explained where samples used for the root data came from. However, I'm still not clear what methods they used to measure root traits. Are the methods from Wambsganss et al? Right now for me line 475 reads as though the soil samples were from that study, but not that the root trait measurements are from that study. Please clarify or provide information regarding methods for root trait measurements.

We have added "see Wambsganss et al., in prep. for root measurement methods" (line 492-493) to clarify that the methods used to measure the absorptive roots are detailed in this article. The root cores were collected concurrently with the soil cores as part of a multi-institutional sampling campaign, and measurements were conducted at the individual institutions.

Line 764-765 (Figure 2). There are two mentions of CO₂ fluxes that should be changed to N₂O fluxes.

Thank you very much for finding this error. It has been corrected (line 790).